



# Reconstruction of flow conditions from 2004 Indian Ocean tsunami deposits at the Phra Thong island using a deep neural network inverse model

**Rimali Mitra[1], Hajime Naruse[1], and Shigehiro Fujino[2]**

[1]Division of Earth and Planetary Sciences, Graduate School of Science, Kyoto University, Kitashirakawa Oiwakecho, Kyoto, 606-8502, Japan
[2]Faculty of Life and Environmental Sciences, University of Tsukuba, 1-1-1 Tennodai, Tsukuba, Ibaraki, 305-8572, Japan

**Correspondence:** Rimali Mitra (mitra.rimali.a75@kyoto-u.jp)

**Abstract.** The 2004 Indian Ocean tsunami caused significant economic losses and a large number of fatalities in the coastal areas. The estimation of tsunami flow conditions using inverse models has become a fundamental aspect of disaster mitigation and management. Here, a case study involving the Phra Thong island, which was affected by the 2004 Indian Ocean tsunami, in Thailand was conducted using inverse modeling that incorporates a deep neural network (DNN). The DNN inverse analysis reconstructed the values of flow conditions such as maximum inundation distance, flow velocity and maximum flow depth, as well as the sediment concentration of five grain-size classes using the thickness and grain-size distribution of the tsunami deposit from the post-tsunami survey around Phra Thong island. The quantification of uncertainty was also reported using the jackknife method. Using other previous models applied to areas in and around Phra Thong island, the predicted flow conditions were compared with the reported observed values and simulated results. The estimated depositional characteristics such as volume per unit area and grain-size distribution were in line with the measured values from the field survey. These qualitative and quantitative comparisons demonstrated that the DNN inverse model is a potential tool for estimating the physical characteristics of modern tsunamis.

## 1 Introduction

On 26 December 2004, a $M_w$ 9.1 earthquake triggered a devastating tsunami that affected the coastal areas and cities adjacent to the Indian Ocean, which resulted in extensive socio-economic damage and numerous fatalities in several countries including Thailand, Indonesia, Sri Lanka, India and Myanmar (Satake et al., 2006; Sinadinovski, 2006; Rossetto et al., 2007; Pari et al., 2008; Satake, 2014; Philibosian et al., 2017). In Thailand, 8300 people lost their lives, with 70 lives and a village of households being lost on the Phra Thong island in Phang-Nga province (Satake et al., 2006; Masaya et al., 2019). The total damage was estimated to amount to around USD 508 million, which equates to 2.2 % of GDP, while the number of deaths was 4225, with the injured and missing cases (Jayasuriya and McCawley, 2010; Suppasri et al., 2012).

An awareness of tsunami disaster prevention is the most essential criterion to reduce socioeconomic losses suffered by countries lying along the coastlines, such as Thailand, Japan, Indonesia, India and Sri Lanka (Lin et al., 2012). Indeed due to the lower tsunami risk and the higher return period of high-magnitude tsunamis (600 years) (Suppasri et al., 2015), the degree of preparedness, for example, effective evacuation techniques, and appropriate awareness are still in the early stage of development in Thailand (Suppasri et al., 2012). Suppasri et al. (2012) reported that the nation has implemented post-tsunami precautionary measures such as the construction of evacuation shelters at a safe height and distance from the coastline along with the evacuation

routes with evacuation regulations, memorial parks, appropriate structural design and land use management which were aimed at dealing with tsunami waves. Meanwhile, a careful building of sea walls and breakwaters has also been suggested for the area.

To propose further regulations for evacuation plan and tsunami hazard mitigation, evaluating the extent of tsunamis with the flow velocity and the maximum height that the tsunamis could reach is important (Pignatelli et al., 2009). However, these flow parameters have not been directly measured, even for the 2004 Indian Ocean tsunami. It has been reported by Satake et al. (2006) that the maximum elevation that a tsunami reached (tsunami height) in Thailand was between 5 and 20 m, and Tsuji et al. (2006) reported a 19.6 m flow height at the Phra Thong island, while Rossetto et al. (2007) reported a peak tsunami height of 11 m and Jankaew et al. (2008) reported a tsunami height of 5 to 12 m in this area. Meanwhile, other flow parameters, such as flow velocity and depth, remain largely unknown. From the video footage of the tsunami, Rossetto et al. (2007) reported a flow velocity of 6–8 m/s at the Khao Lak area and 3–4 m/s at Kamala beach. Other reported flow velocities from Thailand include 4 m/s at Phuket and 9 m/s at Khao Lak (Karlsson et al., 2009; Szczuciński et al., 2012).

It is important to obtain the flow conditions essential to tsunami hazard mitigation in terms of devising future resilient structural measures by investigating tsunami deposits, which provide crucial information on the flow discharge and the extent of the tsunami inundation (Dawson and Shi, 2000; Sugawara and Goto, 2012; Furusato and Tanaka, 2014; Sugawara et al., 2014; Udo et al., 2016; Koiwa et al., 2018; Masaya et al., 2019). It has been suggested that after distinguishing tsunami deposits through their sedimentological characteristics (Morton et al., 2007; Switzer and Jones, 2008; Szczuciński et al., 2012), they can be used to reconstruct tsunami flow conditions (Jaffe and Gelfenbaum, 2007; Smith et al., 2007; Paris et al., 2009; Sugawara and Goto, 2012; Naruse and Abe, 2017; Tang et al., 2018). The preservation of sedimentary bedforms in the sand sheet, capping bedforms, sedimentary structure, texture, and facies models provides the evidence of flow direction and changes in flow energy and hydrodynamic aspects such as flow height and inundation distance (Choowong et al., 2008; Switzer and Jones, 2008; Costa et al., 2011; Szczuciński et al., 2012; Moreira et al., 2017). Other reconstructions of the tsunami flow conditions at Khao Lak were completed using eyewitness reports, aerial videos and photographs, while the extent of the damage was analyzed using field measurements and satellite imagery (Karlsson et al., 2009). In addition, analysis of the sediment geochemistry and the diatom assemblages also provided insights into the flow conditions of the 2004 Indian Ocean tsunami (Sawai et al., 2009; Sakuna et al., 2012; Andrade et al., 2014).

To reconstruct quantitative values of tsunami characteristics from the deposits, various numerical forward and inverse models which incorporate sediment dynamics as well as transport and depositional equations have been established (Li et al., 2012; Sugawara and Goto, 2012; Jaffe et al., 2012; Johnson et al., 2016; Yoshii et al., 2018). Recently, the deep neural network (DNN) inverse model was proposed (Mitra et al., 2020) and was proven to be effective for reconstructing flow conditions via an examination of the deposits of the 2011 Tohoku-oki tsunami. This model also provides some insight into the uncertainty quantification of the estimated flow parameters using the jackknife method. The DNN inverse model predicted the tsunami flow conditions such as maximum inundation distance, flow velocity, maximum flow depth and sediment concentration from the natural tsunami deposits. The reconstructed inundation length was 4045 m, which is close to the original maximum inundation distance of approximately 4020 m; values of run-up flow velocity were 5.4 m/s, which was close to the spatial average of the measurements which ranged from 1.9 to 6.9 m/s; and the estimations of the maximum flow depth was 4.11 m, which was also within the range of the in situ-measured values from the Sendai plain (Mitra et al., 2020). Thus, this model has reasonable potential to estimate the hydraulic conditions from the 2004 Indian Ocean tsunami that were not measured directly.

The Phra Thong island is one of the locations where the tsunami deposits were preserved without a great amount of topographic irregularities with almost no anthropogenic disturbances in the island. The coastlines of the Phra Thong island were severely eroded and retreated by the 2004 tsunami. However, the presence of widespread mangrove forests with other waterborne plant debris helped in the identifications of the extent and direction of the flow (Fujino et al., 2008, 2010). Historically the island is an important location for the study of tsunami deposits, with pre-2004 tsunami deposits preserved in inter-ridge swales and an overall extensive distribution of paleotsunami deposits having been reported (Jankaew et al., 2008; Fujino et al., 2009). In fact, paleotsunami deposits have been identified at the Phra Thong island, Thailand, by several research teams (Jankaew et al., 2008; Fujino et al., 2008; Sawai et al., 2009; Fujino et al., 2010; Brill et al., 2012b; Pham et al., 2017; Gouramanis et al., 2017; Masaya et al., 2019).

Here, we conduct an DNN inverse analysis of the tsunami deposits measured at the Phra Thong island and reconstruct the flow conditions such as the maximum inundation distance, flow velocity, maximum flow depth and sediment concentrations of five grain-size classes. The inverse model was based on the forward model, which was proposed by Naruse and Abe (2017). The forward model calculations were iterated at random initial flow conditions to produce artificial training data sets that represent depositional characteristics such as the spatial distribution of thickness and grain-size composition. Using the artificial training data sets, the DNN was then trained to establish a relation between the depositional characteristics and the flow conditions. The

post-trained DNN model was ready to predict flow conditions from the tsunami deposits after the performance of the trained DNN was verified using test data sets. The 1D cubic interpolation was applied to the field data sets of the Phra Thong island to fit the data set to model grids. Finally, this DNN inverse model was applied to the field data sets from the Phra Thong island, Thailand, to reconstruct the flow conditions of the 2004 Indian Ocean tsunami. Our inverse model was already validated to be effective for 2011 Tohoku-oki tsunami deposits distributed in the Sendai plain (Mitra et al., 2020). In the case of the Phra Thong island, we validated the results by the field measurements of the tsunami flow depth. Also, the estimated thickness and grain-size distribution of tsunami deposits were compared with the actual measurements. Our inverse analysis results could be used for designing future tsunami hazard assessments and disaster mitigation strategies in Thailand.

## 2   Study area

The study area is the Phra Thong island, which is situated off the west coast of Phang-Nga province (north of Phuket island) and the west coast of southern Thailand (Fig. 1a), and is adjacent to the Indian Ocean (). This study investigated the tsunami deposits distributed in the eastern coast of the Phra Thong island, where the topography near the coastline is a flat plain that mainly consists of shore-parallel beach ridges with intervening swales (Brill et al., 2012a). The 2004 Indian Ocean tsunami flooded the area with waves higher than 6 m and an inundation limit of approximately 2 km inland (Tsuji et al., 2006; Fujino et al., 2010). The tsunami left a widespread sand sheet with a thickness of 5–20 cm (Jankaew et al., 2008; Fujino et al., 2010). Meanwhile, the presence of wet, peaty swales helped in the preservation of the tsunami deposits (Jankaew et al., 2008; Fujino et al., 2009; Gouramanis et al., 2017). Given its natural topography with few artificial features, Phra Thong island is a rare case, which is useful for verifying tsunami sediment transport calculations with less uncertainty (Brill, 2012).

Figure 1b shows the location of the Phra Thong island and the adjacent areas in Thailand where the tsunami deposits have been reported. We considered samples from 29 locations along the transect shown in Figs. 1c and 2. The distance from the pre-event coastline to each sampling site was calculated by projecting of the sites to a flow-parallel reference line (Fujino et al., 2010). Tsunami heights of 6.6, 7 and 12 m were reported near the transect where the coast was extensively eroded and had retreated several hundreds of meters (Jankaew et al., 2008; Fujino et al., 2010). The sediment from shallow seafloors were transported and deposited in large volumes of sand sheet deposition widely along the coast, with the deposit largely composed of medium to fine sand. The deposit became thinner and finer in a landward

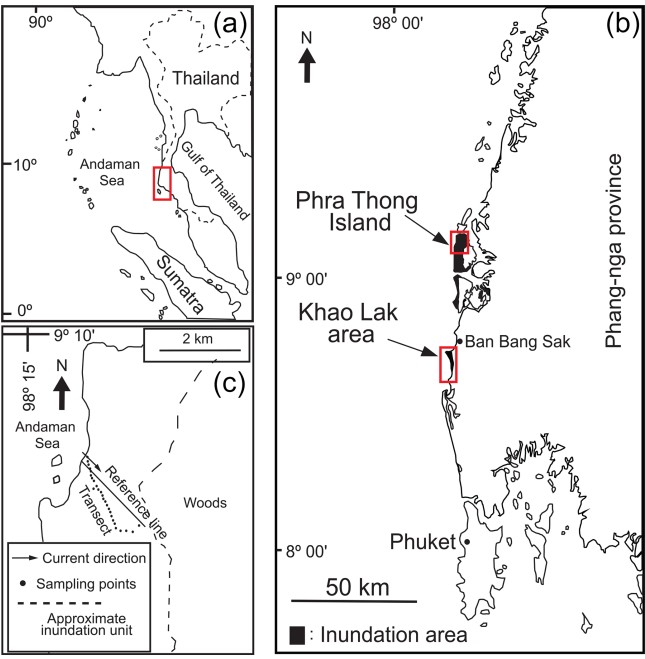

**Figure 1. (a)** Location of study area in southwestern Thailand. **(b)** Phra Thong island and adjacent landmark areas where the 2004 Indian ocean tsunami inundated. **(c)** Locations of study sites at the Phra Thong island. The 2004 tsunami inundated about 2 km inland.

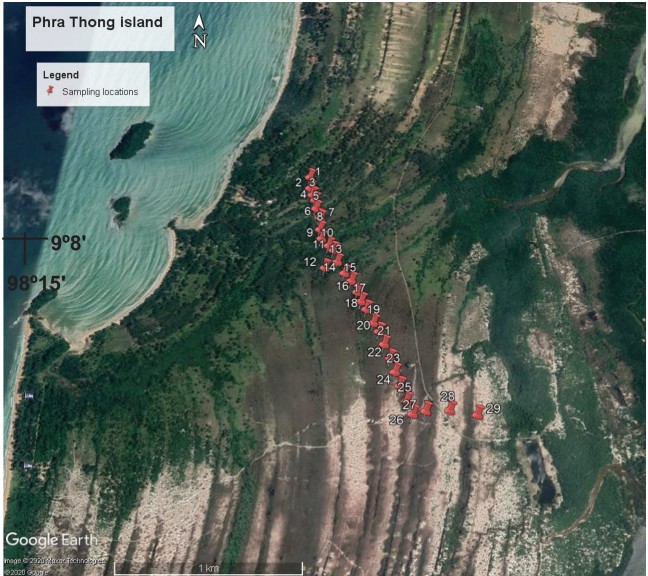

**Figure 2.** © Google Earth image showing locations of sampling points investigated for the 2004 Indian Ocean tsunami of the Phra Thong island described in this paper.

direction, becoming very fine at the landward limit of the inundation.

The maximum inundation distance was measured about 2000 m inland (Fujino et al., 2008, 2010) and the thickness of the tsunami deposits at a maximum of 12 cm, while this did

oscillate a great deal for the first 1300 m from the shoreline. The exponentially landward thinning of the deposits was observed along the transect. For more details on the thickness and grain-size distribution of the tsunami deposit, see the description of the transect of the Phra Thong island provided by Fujino et al. (2010).

The mean grain-size and overall grain-size distribution of the tsunami deposits from the Phra Thong island are shown on Fig. 3b. The overall thickness of the tsunami deposits along the transect are presented in Fig. 3a, and the measured grain-size distributions were discretized to five grain-size classes for every location of sampling sites. It should be noted that along the transect, the initial locations of the sampling points of Fig. 3a and b were adjusted to start from the 0 m distance from the shoreline. Figure 3c and d represent the volume fractions of five grain-size classes and total grain-size distribution.

## 3 Methodology

This model uses the forward model of FITTNUSS (the framework of inversion of tsunami deposits considering transport of nonuniform unsteady suspension and sediment entrainment) (Naruse and Abe, 2017) to calculate the sediment transport and deposition from input parameters including the maximum run-up length, the depth averaged flow velocity, the maximum flow depth and sediment concentration at the seaward end. The forward model can calculate the thickness and grain-size distribution along a 1D shoreline normal transect, which is used to train the DNN inverse model. Here, we present a brief overview of the FITTNUSS forward model and the inverse model.

### 3.1 Forward model

The FITTNUSS forward model is based on the layer-averaged one-dimensional equations that take the following form:

$$\frac{\partial h}{\partial t} + \frac{\partial U h}{\partial x} = 0,\tag{1}$$

$$\frac{\partial U h}{\partial t} + \frac{\partial U^2 h}{\partial x} = ghS - \frac{1}{2}g\frac{\partial h^2}{\partial x} - u_*^2,\tag{2}$$

where $h$ and $U$ denote tsunami flow depth and the layer-averaged flow velocity respectively. The parameters $t$ and $x$ refer to the time and bed-attached streamwise coordinate set which is positive landward and perpendicular to the shoreline, $g$ is the gravitational acceleration, $S$ is the bed slope, and $u_*$ is the friction velocity. Here, we employed the flow resistance law to obtain friction velocity using the friction coefficient, which is widely used in general. A few researchers recently reported that tsunami-induced boundary layers may span only a fraction of water length formula (Lacy et al., 2012; Williams and Fuhrman, 2016; Larsen and Fuhrman,

2019). The importance of the resistance law for the inverse analysis, considering such non-steady conditions, may be a subject for future study. The sediment conservation equation was presented as follows:

$$\frac{\partial C_i h}{\partial t} + \frac{\partial U C_i h}{\partial x} = w_{si}(F_i E_{si} - r_{0i} C_i),\tag{3}$$

where $C_i$ is considered as the volume concentration in the suspension of the $i$th grain-size class and $w_{si}$, $E_{si}$, $r_{0i}$ and $F_i$ are the settling velocity, sediment entrainment coefficient, ratio of near-bed to layer-averaged concentration of the $i$th grain-size class and volumetric fraction of the sediment particles in the bed surface active layer, above the substrate respectively (Hirano, 1971). The details of the parameters and variables are provided in Naruse and Abe (2017).

For the sedimentation of tsunamis, the Exner equation of bed sediment continuity was used, which is expressed as

$$\frac{\partial \eta_i}{\partial t} = \frac{1}{1 - \lambda_p} w_{si}(r_{0i} C_i - F_i E_{si}),\tag{4}$$

where $\eta_i$ refers to the volume per unit area (thickness) of the sediment of the $i$th grain-size class and $\lambda_p$ accounts for the porosity of the bed sediment. As a result of the sedimentation, the grain-size distribution in the active layer varies with time (Hirano, 1971), and the rate of total sedimentation is expressed as follows:

$$\frac{\partial \eta}{\partial t} = \sum \frac{\partial \eta_i}{\partial t}.\tag{5}$$

Finally, using the assumptions proposed by Soulsby et al. (2007), the velocity of the run-up flow of the tsunami, $U$, is assumed as uniform and steady, but the inundation depth varies in time and space. Hence, this model simplification is called the quasi-steady flow assumption (Naruse and Abe, 2017). The flow dynamics of tsunamis were simplified in terms of the following equation:

$$\frac{\partial C_i}{\partial t} + U\frac{\partial C_i}{\partial x} = \frac{R_w}{H(Ut - x)}\{w_{si}(F_i E_{si} - r_{0i} C_i)\}.\tag{6}$$

Here, $R_w$ and $H$ represent the maximum inundation distance and flow depth of the tsunami at the seaward boundary of the transect, respectively. A transformed coordinate system and the implicit Euler method has been applied to the equation to increase the computational efficiency (for more details, see Naruse and Abe, 2017).

Using the above equations, the forward model reproduces the spatial variation of the thickness and grain-size distribution of the tsunami deposit from the input values of the following: (1) maximum distance of horizontal run-up (maximum inundation distance), (2) maximum flow depth, (3) run-up velocity and (4) sediment concentration of each grain-size class at the seaward boundary (Naruse and Abe, 2017). The grain-size classes selected for this inverse analysis were 726, 364, 182, 91 and 46 µm respectively.

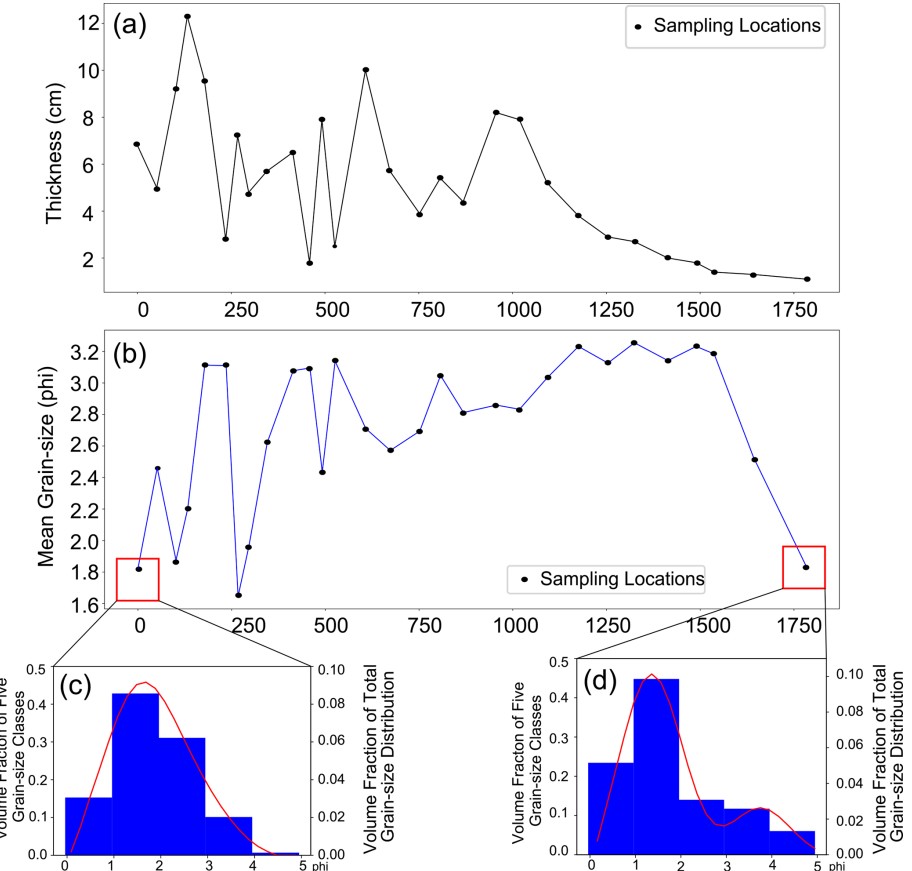

**Figure 3. (a)** Variations of grain-size parameters and thickness of tsunami deposits for the sites along transect of the Phra Thong island. **(b)** Mean grain-size distribution of the tsunami deposits along the transect. **(c, d)** Total grain-size distribution at first and last locations at the Phra Thong island and the discretized fraction of the sediment in the five grain-size classes.

## 3.2 Inverse model

The DNN inverse model (Mitra et al., 2020) accepts grain-size and thickness distribution at an input layer of the neural network (NN). The nodes in the input layers receive the values of the volume per unit area of all grain-size classes at the grid points of the forward model. Then, following the feed-forward mechanism, the NN outputs the tsunami characteristics through the several hidden layers (Fig. 4a) (Mitra et al., 2020). The DNN structure includes the input layer which consists of input nodes where the input values are the volume per unit area of each grain-size class at the spatial grids. Thus, expression of the input nodes numbers is presented as $M \times N$, where $M$ and $N$ are the total number of spatial grids and grain-size classes, respectively. In this inverse model, the total numbers of layers were five, among which the number of hidden layers were three with the 2500 nodes (Mitra et al., 2020). Finally, the output layer consists of the predicted parameters of flow conditions. The details of hyperparameter selection are provided in Mitra et al. (2020).

Before applying the DNN inverse model to the measured tsunami deposits, it was trained using artificial training data

sets of tsunami deposits produced by the repetition of the forward model calculation with randomly generated input values. Figure 4b shows the workflow for training and applying the inverse model. First, the tsunami characteristics values were randomly produced, and the repetition of the forward model calculations using the generated tsunami characteristics produced artificial data sets of the thickness and grain-size distribution of the tsunami deposits to train the NN. The model prediction was evaluated according to the loss function defined as follows:

$$J = \frac{1}{N} \sum \left( I_k^{\text{fm}} - I_k^{\text{NN}} \right)^2, \tag{7}$$

where $I_k^{\text{fm}}$ is denoted as the teaching data that are the initial parameters used for producing in the training data, and $I_k^{\text{NN}}$ denotes the predicted parameters. This loss function quantifies how close the NN was to an ideal inverse model.

The weight coefficients in the NN were optimized to minimize the loss function in the training process (Wu et al., 2018; Mitra et al., 2020). Following the training process, the model could be applied to a measured data set of tsunami deposits. The details of the hyperparameter selection and the

step-by-step procedures of the model training are provided in Mitra et al. (2020).

To generate the training data sets, the present inverse model involves the ranges of input parameters – the maximum inundation distance, maximum flow velocity, maximum flow depth and sediment concentrations of five grain-size classes – for generating the training data sets, which are 1700–4500 m, 2.0–10 m/s, 1.5–12 m and 0 %–2 % respectively. The range of maximum inundation distance can be modified depending on the field evidence of the extent of the tsunami deposit distribution. The range of parameters adopted in this study is applicable to most of the large-scale tsunami-inundated areas as the ranges have been selected with several case studies of tsunamis that include mostly field measurements, survivor video and numerical analysis (Wijetunge, 2006; Fritz et al., 2006; Matsutomi and Okamoto, 2010; Mori et al., 2011; Szczuciński et al., 2012; Abe et al., 2012; Nandasena et al., 2012; Goto et al., 2014).

A sampling window to select the region for applying the inverse model from the entire distribution of the data sets had to be set, given that, in certain cases, the field measurements along the transect do not cover the entire distribution. In addition, the measurements at the distal part of the transect may contain large errors since the tsunami deposits in that area may be too thin for precise observations. The model had to be trained on a specific sampling window, and the precision of the model prediction depending on the sampling window size was tested using the validation data sets. For more details on the significance and applicability of the sampling window, please refer to Mitra et al. (2020).

We have selected a sampling window size of 1700 m for our study, which was chosen on the basis of the comparative results obtained from tests using different sampling window sizes as described in the results section. For this study area, the grid spacing in the fixed coordinates was 15 m, meaning the number of spatial grids used for the inversion was 113.

To apply the inverse model to the measured values of field data set from the Phra Thong island in 1D vectors, the collected data points must be fit into that fixed coordinate system of the model. Here, a 1D cubic interpolation was used on the measured data set that provides values at the positions between the data points of each sample. Since this procedure may have led to additional errors or bias in the results, checking the influence of the interpolation on the predictions of the inverse model using the subsampling of the artificial data sets at the location of the outcrops was essential (Mitra et al., 2020).

The inverse model predicts the flow conditions, and the precision of the results was evaluated using the jackknife method. This method estimates the standard error of the statistics or a parameter of a population of interest from a random sample of data. The jackknife sample is described as the "leave-one-out" resample of the data. If there are $N$ observations, there are $N$ jackknife samples, each of which are $N - 1$. If the sample of $N$ observation is a

set denoted as $x_1, x_2, \ldots, x_N$, the $n$th jackknife sample is $x_1, \ldots, x_{n-1}, x_{n+1}, \ldots, x_N$. The pseudo-value estimation of the $n$th observation was then computed, and an estimate of the standard error from the variance of the pseudo-values was obtained (Abdi and Williams, 2010; Mitra et al., 2020).

## 4 Results

### 4.1 Training and testing of the inverse model

The DNN was trained using artificial data sets of the depositional characteristics such as volume per unit area and grain-size distribution. The number of training data sets was chosen to be 5000 in this study. Figure 6a presents a plot graph of the relationship between the number of training data sets and the loss function of the validation data set. The performance of the inverse model improved as the number of training data sets increased (Fig. 6a), but there was only a slight improvement after the iteration of the forward model calculation exceeded 3000.

The training process proceeded with a certain number of epochs that indicate the iterations of the optimization calculation by the full data set. Figure 6b shows that the present model was reasonably converged over 2000 epochs for both the training and validation performances. The loss function values of training and validation at the first epoch were 0.08 and 0.05, respectively. The final and lowest loss function values at the final epoch were 0.0036 for the training data sets and 0.0013 for the validation data sets.

After training the model, the predictions of the inverse model for the test data sets were plotted against the original values used for producing the data sets. Figure 7a–h show that the eight predicted parameters from the artificial test data sets were distributed along the 1 : 1 line in the graph, indicating that the test results were correlated well with the original inputs. Figure 8a–h show the histograms of the deviation of the estimated values predicted from the original values. Deviations were distributed in a relatively narrow range without large biases in relation to the true conditions, except in the case of the maximum flow depth, which was slightly biased. The values of the predicted maximum flow depth were approximately 0.43 m lower than the input values.

### 4.2 Application of the DNN inverse model to the 2004 Indian Ocean tsunami

#### 4.2.1 Inversion results

The inversion method was applied to the measured grain-size distribution of tsunami deposits along the transect of the Phra Thong island in view of reconstructing the flow conditions from the deposit of the 2004 Indian Ocean tsunami. The 1D cubic interpolation was applied to the data set measured along the transect of the Phra Thong island, before the inversion method was applied to the field data set.

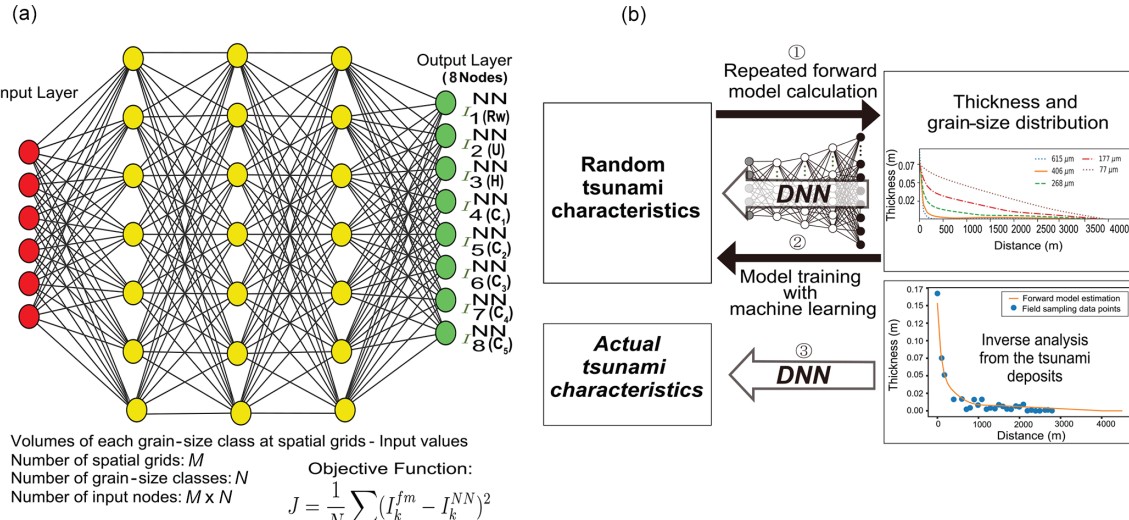

**Figure 4. (a)** NN architecture of the DNN which predicts the output maximum inundation distance ($R_w$), flow velocity ($U$), maximum flow depth ($H$) and concentration of five grain-size classes ($C_1$ to $C_5$) (modified from Mitra et al., 2020). **(b)** Flow chart of the inverse model (modified from Mitra et al., 2020).

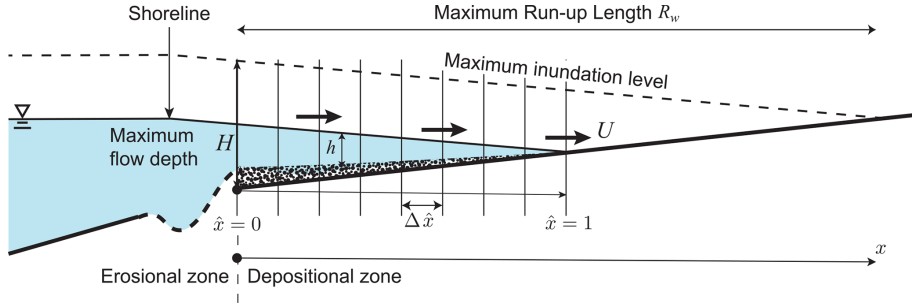

**Figure 5.** Explanation of model domain configuration. The assumption of velocity of tsunami run-up $U$ is constant in time and space. The inundation depth $h$ increases constantly until it reaches its maximum value $H$ at the seaward boundary. $R_w$ is the maximum inundation distance. The bed-attached streamwise coordinate $x$ is set transverse to the shoreline and is positive landward. Within the applied transformed coordinate system, the moving front edge of the tsunami is located at a fixed value of the dimensionless spatial coordinate $\hat{x} = 1$ (modified from Naruse and Abe, 2017).

We selected 1700 m as the length of the sampling window, which allowed for minimizing the uncertainty of the inverse analysis quantified via the jackknife method (Fig. 9). The jackknife standard error was calculated for different sampling window sizes of the data sets. Figure 9 shows that the error decreased as the sampling window was increased, with the exception of the region above 1700 m. However, an increasing trend was observed for maximum flow depth, while the jackknife standard error became stable after 1500 m (Fig. 9c). Thus, the 1700 m sampling window provided the best results in terms of the precision of the inversion. As described in the method section, the interpolation of the measured data sets at the computational grids may result in additional bias or errors from the inverse model. The subsampling analysis was thus conducted using artificial data sets. This test was done to check the effect of irregularly spaced

field data sets on the accuracy of the inversion. The details on the subsampling procedure are given in Mitra et al. (2020).

The subsampling test demonstrated that the inversion model had a mean bias of 10.82 m for maximum inundation distance (Fig. 10), while the predicted result by DNN was 1700 m. Likewise the predicted results for the flow velocity was 4.63 m/s, and it was 4.82 m for the maximum flow depth, with the mean bias obtained from the subsampling results being 0.14 m/s for flow velocity and $-0.43$ m for maximum flow depth, which were exactly in line with the values obtained from the testing of the trained DNN model without the subsampling test.

Table 1 shows the predicted flow conditions with a 95 % confidence interval calculated by jackknife method (Fig. 12). When using the jackknife standard error calculations, the maximum inundation distance was 1700 m with a 8.09 m

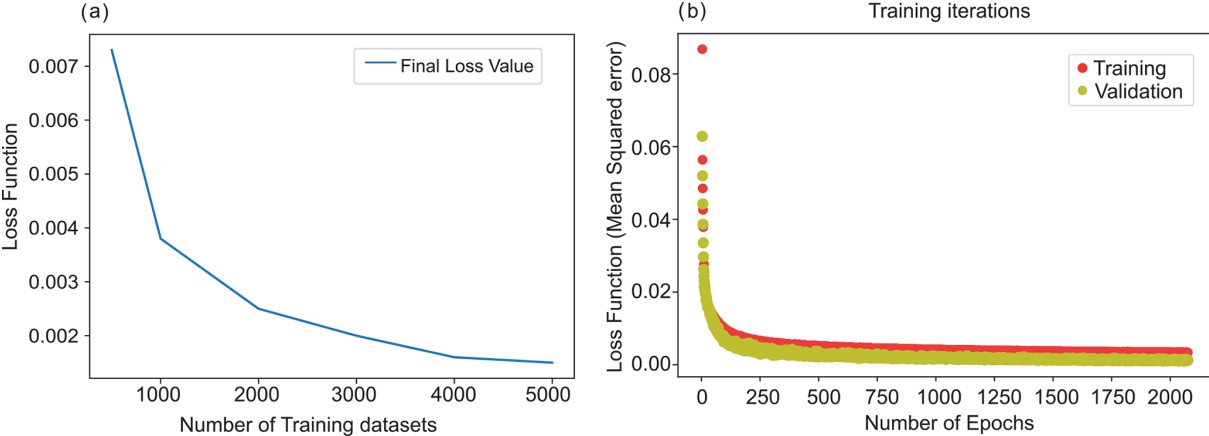

**Figure 6. (a)** Relationship between the loss function of the validation and the number of training data sets selected for the inverse model. The results of the training improved as the number of training data sets increased, while it slightly varied after 5000 training data sets. **(b)** History of learning indicated by the variation of the loss function (mean squared error). Both values of the loss function for the training and validation data sets reached a minimum value, indicating that overlearning did not occur.

**Table 1.** Predicted results from the inverse model when applied to the 2004 Indian Ocean tsunami data obtained from the Phra Thong island, Thailand. All reported standard error calculations were performed using a 95 % confidence interval.

| Parameters | Predicted results | Mean bias |
|---|---|---|
| Maximum inundation distance | 1700 m $\pm$ 8.09 m | 10.82 m |
| Flow velocity | 4.63 m/s $\pm$ 0.20 m/s | 0.14 m/s |
| Maximum flow depth | 4.82 m $\pm$ 0.25 m | $-0.43$ m |
| Concentration of $C_1$ (726 µm) | 0.17 % $\pm$ 0.017 % | 0.01 % |
| Concentration of $C_2$ (364 µm) | 0.22 % $\pm$ 0.017 % | 0.008 % |
| Concentration of $C_3$ (182 µm) | 0.17 % $\pm$ 0.032 % | $-3 \times 10^{-4}$ % |
| Concentration of $C_4$ (91 µm) | 0.27 % $\pm$ 0.010 % | 0.007 % |
| Concentration of $C_5$ (46 µm) | 0.01 % $\pm$ 0.001 % | 0.008 % |

range of uncertainty (Fig. 12a). Meanwhile, the estimated flow velocity was 4.63 m/s, and the maximum flow depth was 4.82 m, with jackknife standard error uncertainty values of 0.20 m/s and 0.25 m, respectively (Fig. 12b–c). The reconstructed total sediment concentration over five grain-size classes was approximately 0.8 %, and the estimated values of each grain-size class ranged from 0.01 %–0.27 %. The jackknife error estimation shows the presence of errors was low, such as 0.001 % (Table 1).

Finally, the forward model calculation was performed using the reconstructed flow conditions to estimate the spatial distribution of the volume per unit area and grain-size composition, and it was compared with the measured values from the transect of the Phra Thong island. Figure 11 shows the predicted spatial grain-size distribution was in line with the actual values from field measurements.

## 5   Discussion

### 5.1   The model's inversion performance

The training and testing of the DNN inverse model demonstrated that this model has reasonable ability to predict tsunami characteristics such as maximum inundation distance, flow velocity, maximum flow depth and sediment concentrations. The final loss function values for the training and validation were 0.0036 and 0.0013 respectively, which were close (0.0040 and 0.0018) to those reported by Mitra et al. (2020). The testing of the DNN inverse model was evaluated using artificial data sets of tsunami deposits. The scatter diagrams (Fig. 7) of the predicted and true conditions indicate a good correlation, with no large deviation in the mode of the predicted values except for a slight bias in the maximum flow depth. While the model tended to estimate the maximum flow depth values approximately 0.43 m lower on average, correcting the final results by adding the bias to the final reconstructed values from the original field data was possible. In Mitra et al. (2020), the reported bias for the maximum flow depth was approximately 0.5 m, while the sample standard deviation was around 0.40, which is close to the value in the present study (0.38 m). The bias during the testing of the model was caused by the internal algorithm and neural network structure, but we hope the bias of maximum flow depth will be sorted if we improve the neural network structure in the future. In future studies, the algorithm of the neural network structure can be improved to eliminate or reduce the bias of the parameter.

Regarding the deviation of the predicted values from the true values which are artificial test data sets, the sample standard deviation values were relatively small for all parameters. The sample standard deviation for the maximum inundation distance was as low as 88.70 m for a range of true values

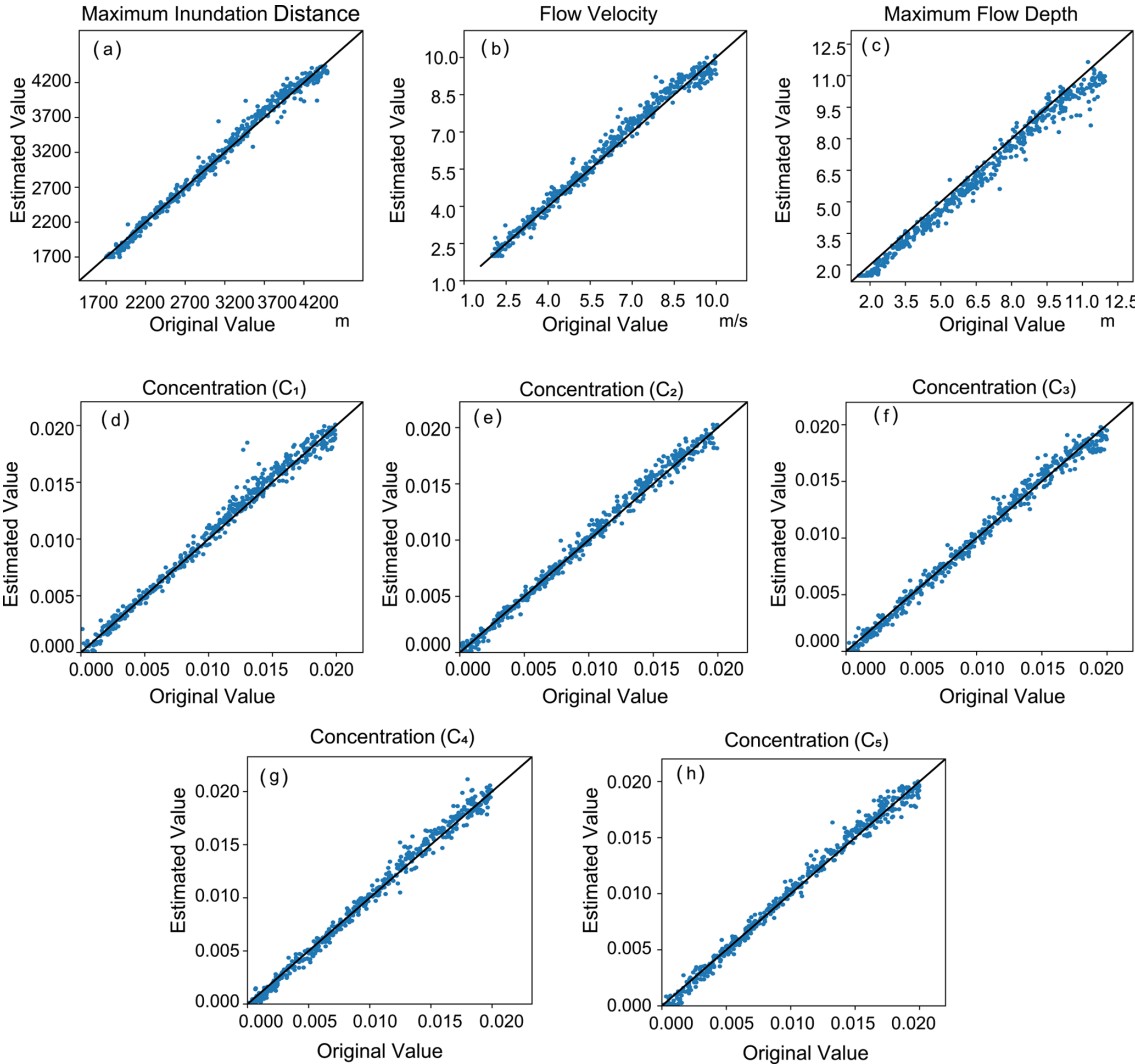

**Figure 7.** Performance verification of the model using artificial test data sets, indicating that the values estimated using the inverse model were plotted against the original values used for the production of the test data sets. Solid lines indicate a 1 : 1 relation and suggest good correlation.

of 1700–4500 m, while that for flow velocity was 0.29 m/s for a range of true values of 2.0–10 m/s. Meanwhile, the average value for sediment concentration was around 0.05 %. All these values were close to those reported by Mitra et al. (2020) (e.g., maximum inundation distance, 77.03 m; flow velocity, 0.30 m/s; sediment concentration, 0.06 %).

After the model was trained and tested, the test data sets were subsampled at the sampling locations on the Phra Thong island to investigate the bias in the predicted flow conditions due to the irregular distribution of the sampling points. The results implied that the irregularity of the sampling distribution had little effect on the bias and errors. In fact, the bias values for maximum inundation distance, flow velocity, and sediment concentration were very small (Fig. 10a–e), while that for the maximum flow depth in the subsampling tests indicated no additional bias, implying that

the sampling interval on the Phra Thong island was sufficient for the inverse analysis using the DNN model.

To summarize, the performance of the trained DNN inverse model was identical to that of the model reported in Mitra et al. (2020), which successfully reconstructed various characteristics of 2011 the Tohoku-oki tsunami. It is noteworthy that Mitra et al. (2020) used a different number of grain-size classes than used in our study, and they also employed different ranges of initial parameters for flow velocity and maximum inundation distance. The modifications in the current study were necessary since the grain-size distribution of the tsunami deposits measured at the Phra Thong island was considerably coarser than that measured in the Sendai plain. This change had close to zero effect on the performance of the inverse model, implying that the inverse method employed in this study is adaptable to various environments.

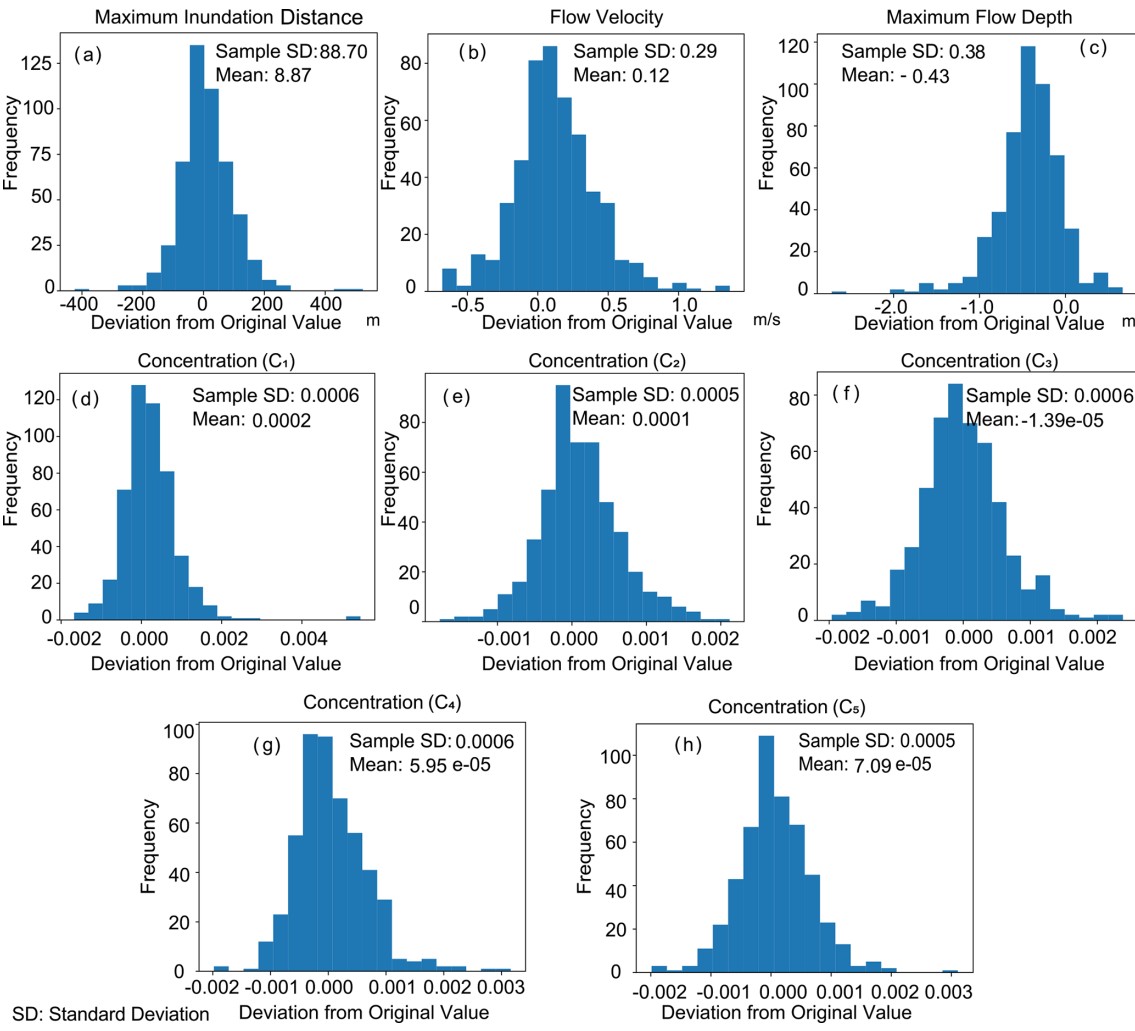

**Figure 8.** Histograms showing the deviation of the predicted results from the original values of the artificial test data sets.

## 5.2 Verification of inversion results for the tsunami deposits

After the testing of the inverse model described above, we applied the model to the data sets obtained along the transect (Fig. 1) and obtained the first quantitative estimates of the tsunami characteristics in the Phra Thong island. While in situ measurements of the 2004 Indian Ocean tsunami's activity on in the Phra Thong island are not abundant, several surveys have reported the attendant inundation heights and run-up length of the tsunami in this region. Here we compare our inversion results with these in situ measurements of the 2004 Indian Ocean tsunami.

The inversion results or the tsunami flow depth in this study were in the range of the in situ measurements. The DNN inverse model reconstructed the maximum flow depth as $4.82 \pm 0.25$ m at the sampling site, which was located approximately 684 m from the shoreline, when measured in the direction parallel to the flow direction (N154° E). This value

does not contain the additional bias $-0.43$ m. The data of tsunami inundation height, which present a sum of the flow depth and topographic height, were measured at the Phra Thong island by several research groups including Tsuji et al. (2006) and the Korean Society of Coastal and Ocean Engineers (KSCOE) groups (Choi et al., 2006) (http://www.nda.ac.jp/~fujima/TMD/fujicom.html, last access: 5 December 2021). The data points reported by the latter were 50 and 400 m from the shoreline and were relatively close to sampling site 1 (distances of approximately 1.40 and 1.37 km away from sampling site 1). The measured values of the tsunami inundation heights at these sites were 7.1 and 6.7 m. The KSCOE group also reported the inundation heights at four sites in the Phra Thong island, which were 30–130 m from the shoreline and relatively far from the transect (ca. 2.55 km from sampling site 1), with the inundation heights found to be between 5.5–6.0 m at these sites. Meanwhile, the averaged elevation around the study area which was calculated from the topographic profiles pro-

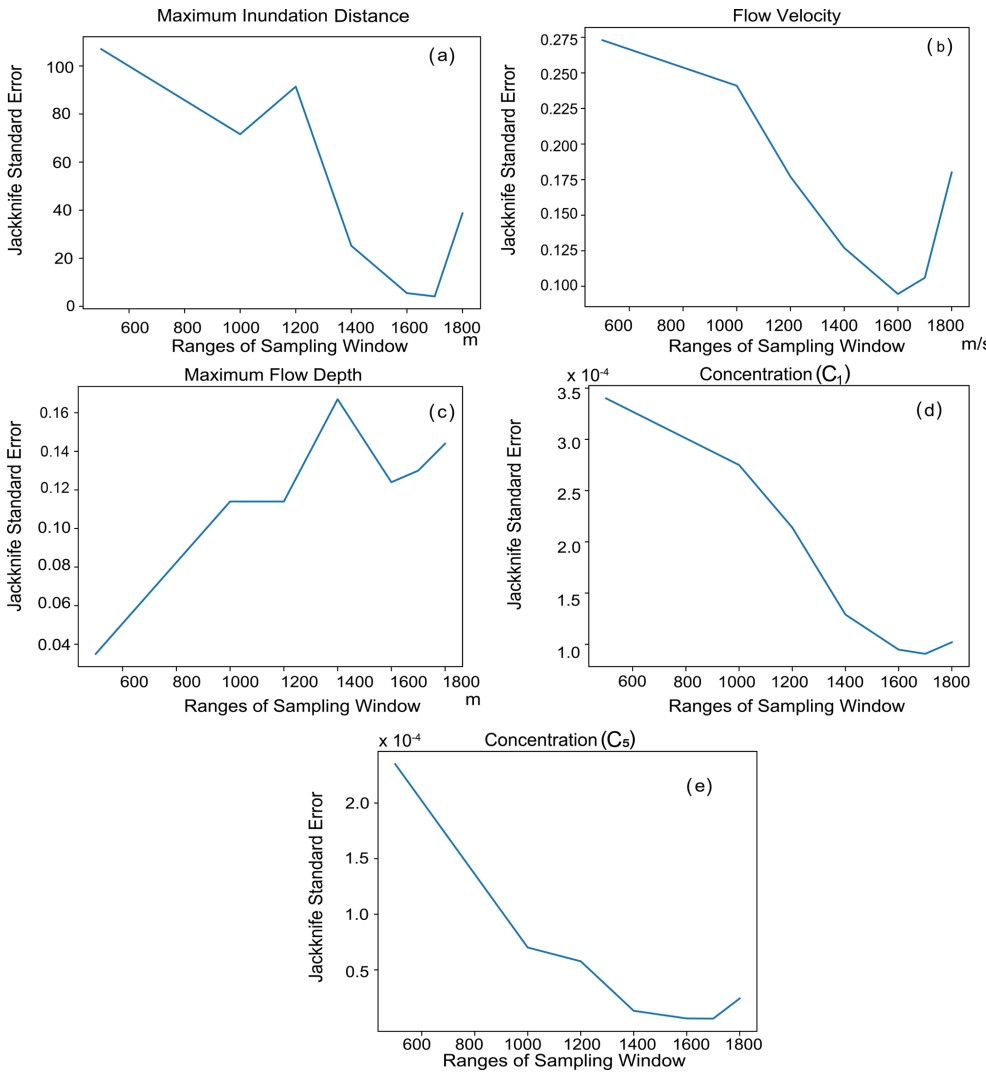

**Figure 9.** Propagation of jackknife standard errors with a different range of sampling window distances.

vided by Jankaew et al. (2008), Jankaew et al. (2011) and Brill et al. (2012b) was approximately 2.9 m. The most seaward locations of the transect in Jankaew et al. (2011) and Jankaew et al. (2008) were around 400 m from sampling site 1 in our study area. The maximum and measured flow heights from the Phra Thong island were reported to be 7.1 and 5.5 m respectively (http://www.nda.ac.jp/~fujima/TMD/fujicom.html). The corresponding maximum and minimum values of elevation are 3.1 and 1.1 m respectively (Jankaew et al., 2008, 2011; Brill et al., 2012b). Hence, the approximate estimate of measured maximum flow depth ranged from 2.4 to 6.0 m. Considering the bias correction of 0.43 m, the reconstructed value of maximum flow depth (5.3 m) falls within the range of measured maximum flow depth values. Hence, when based on the 1700 m sampling window size, the maximum flow depth reconstructed in this study was close to the reported measurements. However, a certain amount

of measurement and calculation error may have existed due to the local topographical variations. The model also estimated a maximum inundation distance (1700 m) that was close to the observed value of approximately 2000 m, which was measured at the inland end of the transect (Fujino et al., 2010).

## 5.3 Characteristics of the 2004 Indian Ocean tsunami on the Phra Thong island

Our inversion results for the tsunami characteristics on the Phra Thong island indicated that the tsunami inundation flow was typically uniform along the coastal area of Thailand. This study reconstructed the flow velocity of the tsunami as $4.63 \pm 0.20$ m/s. Given that no direct observation values have been reported for this specific transect in the Phra Thong island, this presented the first estimate for this region. The reconstructed flow velocity in this region was close to the ob-

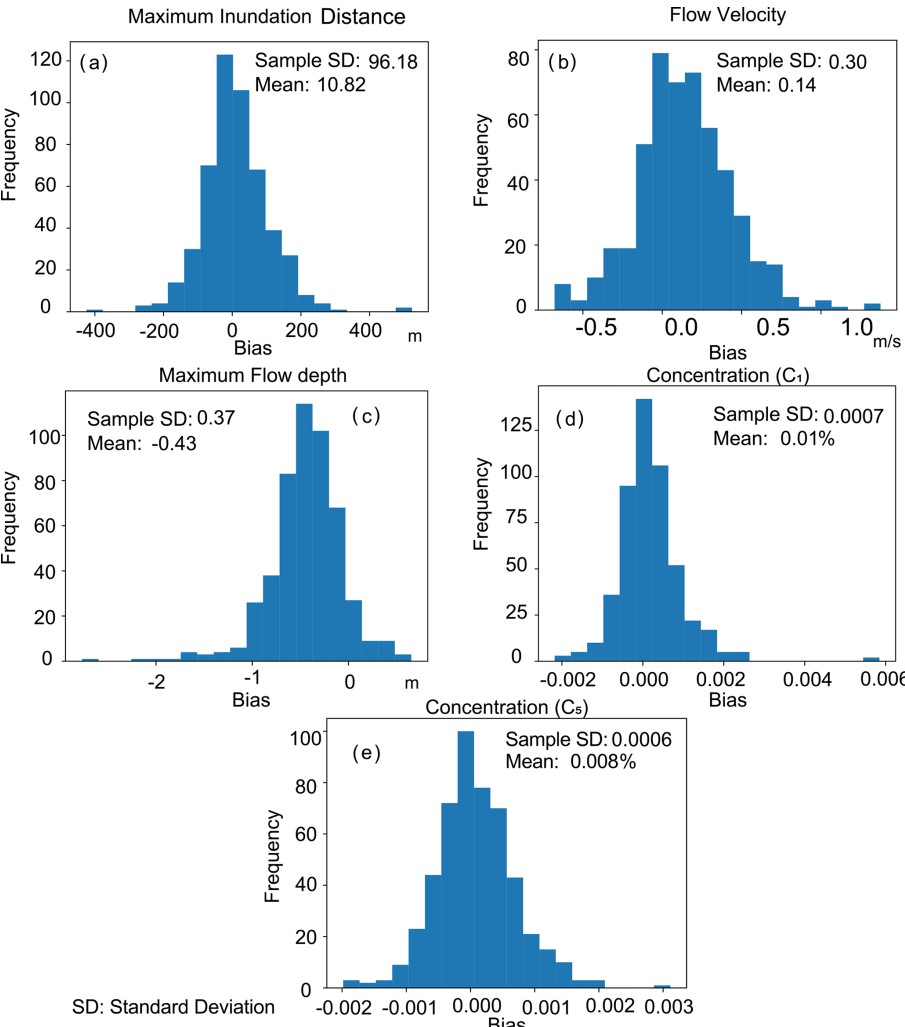

**Figure 10.** Histograms showing the variance and bias of predictions from the test data sets subsampled at the sampling locations of the transect in the Phra Thong island.

served velocity in other regions of coastal areas in Thailand, although a larger velocity was reported in the Khao Lak area. Rossetto et al. (2007) reported video footage of the flow velocity, which was around 3–4 m/s on Phuket island (118 km south of our study area) and 6–8 m/s in the Khao Lak area (43 km south of our study area). Given the values collected from the video footage (Rossetto et al., 2007) in relation to Phuket island, the Khao Lak area and the results reported by Brill et al. (2014), it is clear that most of the flow velocity values were around 4–5 m/s, apart from the Khao Lak area. In fact, the flow depth measurement data from the Khao Lak area also had exceptionally high values (Tsuji et al., 2006; Karlsson et al., 2009), indicating that the tsunami inundation flow could have been locally enhanced by the topographic effects in this region. The flow velocity and depth of the 2004 Indian Ocean tsunami were similar in all other regions covering a 130 km area from Phuket to Phra Thong island.

## 5.4 Comparison with the results of the existing 2D forward model

While the inverse analysis of tsunami deposits provides estimates of the flow characteristics in specific regions, two- or three-dimensional forward modeling is required to infer the spatial distribution of the flow parameters on a regional scale (Li et al., 2012; Masaya et al., 2019). The horizontal two-dimensional forward model TUNAMI-N2 was applied to the Phra Thong island to estimate the spatial distribution of the maximum flow depth in this area (Masaya et al., 2019). However, model appeared to have overestimated the maximum flow depth when compared with the measured values obtained by the KSCOE group (Choi et al., 2006), with the former returning a flow depth of 6–8 m and the latter returning a depth of 4.2–3.8 m. This model is based on a fixed-source model where the initial water levels for a whole region are set along with the specific fault parameters. The model's results

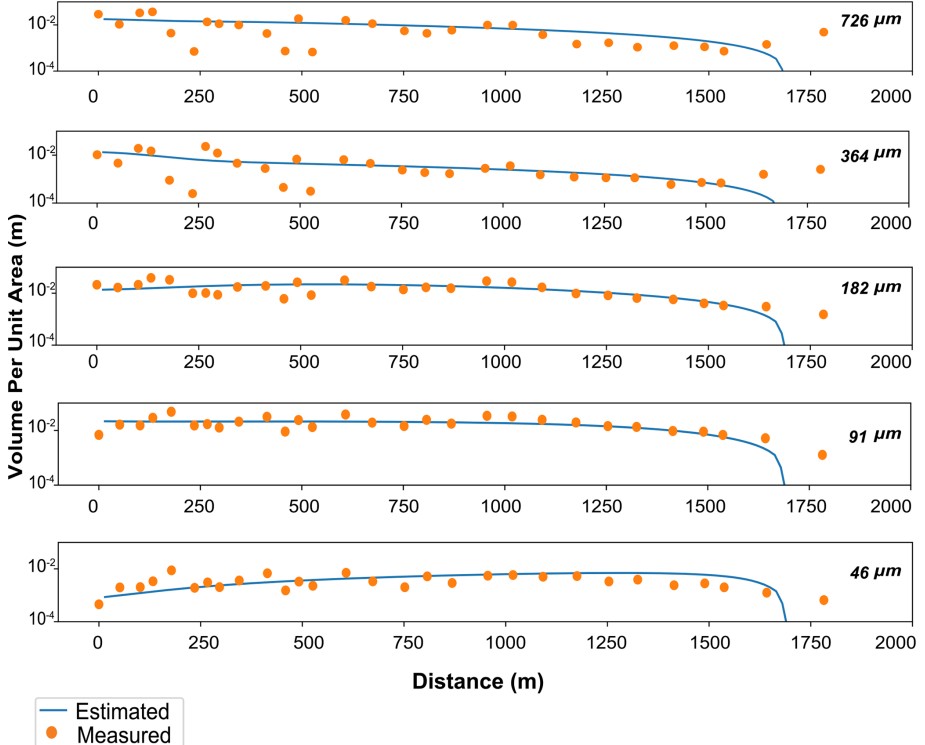

**Figure 11.** Spatial distribution of volume per unit area of five grain-size classes. Solid circles indicate the values measured by Fujino et al. (2010), and lines indicate the results of the forward model calculation obtained using parameters predicted by the DNN inverse model.

strongly depend on these fault parameters, which should be iteratively modified to fit the measurement or distribution of the actual tsunami deposits. In addition to the source model, this model also includes tsunami sediment transport calculation that consists of bed load layer and suspended load layer. However, the calculated value of the sediment thickness was overestimated as the assumption of a movable bed for a large area caused excessive erosion of the ground (Masaya et al., 2019). Moreover, the model of Masaya et al. (2019) employed a single grain-size class for the reconstruction of the parameters from a larger area, which could have resulted in an erroneous estimation as the distribution of grain size of tsunami deposits varies due to sediment transportation and deposition (Sugawara et al., 2014). In contrast, the DNN inverse model does not involve predefined conditions or thresholds to deduce the maximum flow depth. Here, the estimated flow characteristics and thickness distribution of the deposits by the DNN inverse model fit well with the measured values, but they only apply to a local region. However, the DNN inverse model can potentially accept any type of forward models that can produce the distribution of tsunami deposits as training data sets. The model calculation of Masaya et al. (2019) relies on the estimation of a single set of fault parameters, which were not widely explored to obtain the optimal parameters. In the future, the TUNAMI-N2 model can be potentially used as the forward model in the DNN inverse

model to consider the two-dimensional behavior of tsunamis. To do so, the model needs to be modified to consider the sediment transport of multiple grain-size classes.

## 6 Conclusions

The DNN inverse model demonstrated its efficiency in successfully reconstructing the hydraulic conditions of the 2004 Indian Ocean tsunami from the Phra Thong island, Thailand. The reconstructed maximum inundation distance was 1700 m, while the flow velocity and maximum flow depth were 4.63 m/s and 4.82 m respectively. The value of maximum flow depth including the additional bias correction was 5.3 m, which was within the range of 2.4 to 6.0 m, which was the approximate estimate of measured maximum flow depth at the Phra Thong island. The value of flow velocity was also close to the reported values using the video footage from the vicinity of the Phra Thong island. The uncertainty of the results using jackknife method also indicated that simulated results did not contain a large range of values. Phra Thong island was one of the most well preserved and historically important areas for paleotsunami deposits. Hence, the application of the DNN inverse model was suitable to reconstruct flow conditions of the 2004 Indian Ocean tsunami from the Phra Thong island. The DNN inverse model also represented the comparison of the calculated and measured spatial distri-

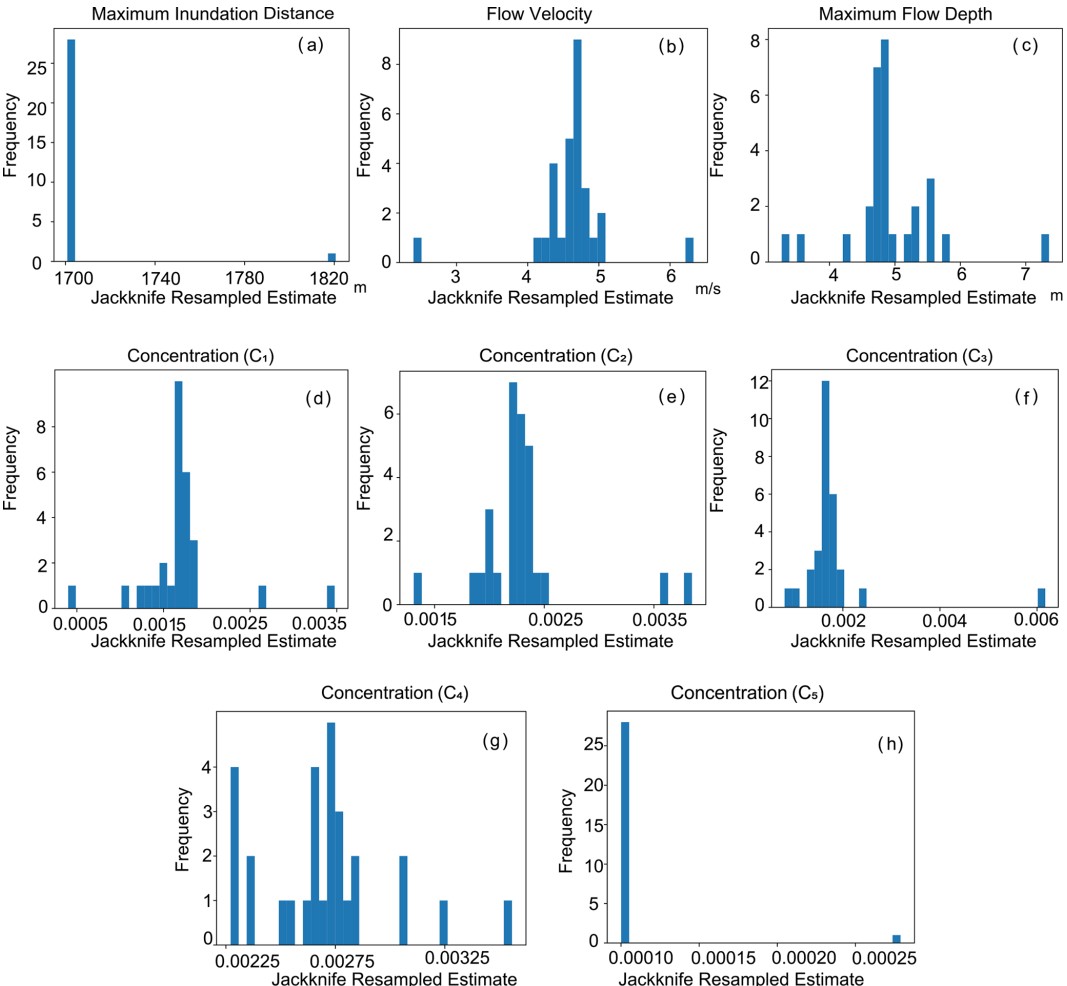

**Figure 12.** Jackknife estimates for the results predicted by the inverse model at the 1700 m sampling window, used to determine the uncertainty of the model.

bution of volume per unit area along the transect at the island. This model can be applied to any areas of modern and ancient tsunami deposits consisting of low land or flat areas to successfully reconstruct the tsunami flow conditions and can serve as a tool for tsunami hazard assessment and disaster resilience at coastal cities.

*Code and data availability.* The source codes and all other data of the DNN inverse model are available in Zenodo (https://doi.org/10.5281/zenodo.4744889, Mitra et al., 2021).

*Author contributions.* HN designed the research; HN and RM performed the research; SF contributed the data from the Thailand area and analyzed the grain-size distribution; RM and HN wrote the paper.

*Competing interests.* The authors declare that they have no conflict of interest.

*Acknowledgements.* We thank the funding providers and the Ministry of Education, Culture, Sports, Science and Technology, Japan, for providing the permission and scholarship for conducting this collaborative research in Japan. We are thankful to the editor Maria Ana Baptista, Pedro Costa and two anonymous reviewers for their detailed and constructive suggestions that significantly improved the paper.

*Financial support.* This research has been supported by the JSPS KAKENHI Grants-in-Aid for Scientific Research (B) (grant no. 20H01985) and the Sediment Dynamics Research Consortium (grant no. 200180500003).

*Review statement.* This paper was edited by Maria Ana Baptista and reviewed by Pedro Costa and two anonymous referees.

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
