# Peer review of "Reconstruction of flow conditions from 2004 Indian Ocean tsunami deposits at the Phra Thong island using a deep neural network inverse model"

_Natural Hazards and Earth System Sciences, 2020_

## Referee Comment (RC1) · Pedro Costa (Referee) · 24 Nov 2020

Dear authors, thanks for the opportunity to review your work on Reconstruction of flow conditions from 2004 Indian Ocean tsunami deposits at the Phra Thong island using a deep neural network inverse model.

I deeply appreciated your effort and I consider that this manuscript has potential to be accepted for publication at NHESS after some major revisions. I detailed them on over 100 comments on the annotated version attached.

[Figure]

I find the aim of the manuscript really interesting and at the forefront of tsunami geoscience. To couple forward and inverse modelling to reconstruct tsunami flow characteristics is essential for the scientific community to be able to provide accurate reconstructions of past events that can be passed to coastal planners and civil protection authorities to better plan adaptation, mitigation and emergency plans.

The numerical modelling you applied has recently proved to be reliable (Mitra e al., 2020, JGR). Therefore, I was expecting some major steps forward in this study case instead of just another small incremental exercise. The Science is there but English must be revised extensively and above all there must be a clear clarification on how and what exactly field data was used to validate the model. I believe you used Fujino et al. (2010) data but sometimes when reading the manuscript one feels puzzled to confirm that you did use it and which values have you used it. For instance, thickness, grain-size curve, etc. Therefore, I cannot agree with the title proposed because it was not well-establish that the regressive model used geological data. Sometimes the reader feels, the models fed and validated each other and no solid, extensive and accurate field data was used. I assumed that this might just be a language and writing problem. Even if it is that, you need to address it. Sometimes the text is confusing and one wonders what you trying to transmit. For example, when you state the model was validated by "observed" flow depths in several locations along the studied profile...in fact, you are saying that the model agrees well with previous modelling exercises for flow depth establishment. The meaning of both sentences in totally different regarding field validation and this is crucial for this manuscript.

There are many other aspects I raised on the annotated version and I suggest you analyse them critically. I might have misunderstood some wording (which means that you need to make it clear) or I might have perceived things correctly (which means you need to change the structure and scope of the manuscript). One example, is sediment concentration. How can you validate flow sediment concentration from the deposit? Only if you look at grain-size curve, spatial distribution and packing (inner

architectural arrangement) of the deposit. You never mention this along the manuscript which means that I am puzzled how you reconstruct sediment concentration on the inverse model. It is easy to understand how you do it with the forward model but departing from sediments (without mentioning the characteristics above) is baffling.

Having said this, the manuscript has lots of potential and after a detailed revision might be accepted for publication on NHESS. I acknowledge the authors effort and hope they take my comments constructively. I believe most of my comments here and especially on the annotated version will improve your manuscript. Kind regards Pedro JM Costa

———————————————————

[Figure]

[Figure]

**Reconstruction of flow conditions from 2004 Indian Ocean tsunami deposits at the Phra Thong island using a deep neural network inverse model**

Rimali Mitra[1], Hajime Naruse[1], and Shigehiro Fujino[2]

[1]Division of Earth and Planetary Sciences, Graduate School of Science, Kyoto University, Kitashirakawa Oiwakecho, Kyoto, 606-8502, Japan.
[2]Faculty of Life and Environmental Sciences, University of Tsukuba, 1-1-1 Tennodai, Tsukuba, Ibaraki, 305-8572, Japan

**Correspondence:** Rimali Mitra (mitra.rimali.37z@st.kyoto-u.ac.jp)

**Abstract.** The 2004 Indian Ocean tsunami caused major topographic changes that resulted in significant economic losses and a large number of fatalities in the coastal areas. The estimation of tsunami flow conditions using inverse models has become a fundamental aspect of disaster mitigation and management. Here, in relation to the 2004 Indian Ocean tsunami, a case study involving the Phra Thong island in Thailand was conducted using inverse modeling that incorporates a deep neural network (DNN). The inverse analysis reconstructed the values of flow conditions such as maximum inundation length, flow velocity and maximum flow depth, sediment concentration from the post-tsunami survey around Phra Thong island. The quantification of uncertainty was also reported using the jackknife method. Using other models applied to areas in and around Phra Thong island, the predicted flow conditions were compared with the reported observed values and simulated results. The estimated depositional characteristics such as volume per unit area and grain-size distribution, were in line with the measured values from the field survey. These qualitative and quantitative comparisons demonstrated that the DNN inverse model is a potential tool for estimating the characteristics of modern tsunamis.

**1 Introduction**

On December 26, 2004, a Mw 9.1 earthquake triggered a devastating tsunami that affected the coastal areas and cities adjacent to the Indian Ocean, which resulted in extensive socio-economic damage and numerous fatalities in several countries including Thailand, Indonesia, Srilanka, India, Myanmar (Rossetto et al., 2007; Satake et al., 2006; Sinadinovski, 2006; Philibosian et al., 2017; Satake, 2014; Pari et al., 2008). In Thailand, 8300 people lost their lives, with 70 lives and a village of households were lost on the Phra Thong island in Phang-Nga province (Satake et al., 2006; Masaya et al., 2019). The total damage was estimated to amount to around USD 508 million, which equates to 2.2% of GDP which while the number of deaths was 4225, with the injured and missing cases and the cost reconstructing properties much lower than the overall damage value (Jayasuriya and McCawley, 2010; Suppasri et al., 2012).

An awareness of tsunami disaster prevention is the most essential criterion to reduce socioeconomic losses suffered by countries lying along the coastlines, such as Thailand, Japan, Indonesia, India and Srilanka etc (Lin et al., 2012). However, it

**Fig. 1.**

---

## Referee Comment (RC2) · Pedro Costa (Referee) · 24 Nov 2020

The comment was uploaded in the form of a supplement:
https://nhess.copernicus.org/preprints/nhess-2020-373/nhess-2020-373-RC2-supplement.pdf

---

## Referee Comment (RC3) · Anonymous Referee #2 · 26 Nov 2020

The paper presents reconstruction of flow conditions from the 2004 Indial Ocean tsunami deposits at the Phra Thong islant, based on a deep neural network iverse model. It bears strong similarity to the techniques developed in their 2020 paper (Mitra et al. 2020, JGR Earth Surf., DOI: 10.1029/2020JF005583, deemed M2020 below), and hence the methodology would not seem to be new. However, the paper does demonstrate novelty as a similar application at a new site (Phra Thong island in this work vs. Sendai plain in the JGR paper), thus establishing greater applicability. The work is presented reasonably, but in some instances I feel should require some further details. I recommend eventual publication, provided the following issues can be addressed.

The figures presented as Figure 1 in this paper have already appeared in M2020 (their Figs. 1 and 2). This is at least acknowledged in the caption. Figure 2 also appears identical to Fig. 3 of M2020 (apart from some color changes), but this does not seem to be indicated in the Figure caption. Is this indeed essentially the identical figure? If so, this ought to be acknowledged. (It could just be very similar, and the differences not perceptable).

Several details of the numerical model used to generate test data sets in Section 2.1 are seemingly missing. These include, the following issues: How are the friction velocity ($u\_*$), the setting velocity ($w\_{s,i}$), the sediment entrainment coefficient ($E\_{si}$), and other variables ($r\_{0i}$ and $F\_i$) determined?

Regarding the friction velocity ($u\_*$), in particular, it should be noted that great care ought to be taken for this quantity if standard (based on steady flow) friction formulas are used, as several recent research papers have shown that tsunami induced bounday layers may span only a fraction of the water depth, and hence these may well be invalid. See e.g. Lacy et al. (DOI: 10.1029/2012JC007954), Williams & Fuhrman (DOI: 10.1016/j.coastaleng.2015.12.002), Tinh & Tanaka (DOI: 10.1080/21664250.2019.1672127) or Larsen & Fuhrman (DOI: 10.1016/j.coastaleng.2019.04.011). Please clarify this point, and if this is indeed being done, this potential defficiency ought to at least be acknowledged.

Can a definition sketch of the model domain (etc.) being used for the generation of the training data sets please be provided? This will help readers immensely to get an idea as to the actual setup being used. Plots just showing performance (like Figs. 2 and 3) fail to provide this.

I do not find that the DNN architecture being used is presented with sufficient clarity. In Section 2.2 (top) it is stated that the DNN model accepts grain-size and thickness distribution at an input layer, and that the outputs are the "tsunami characteristics through several hidden layers". This is rather unclear. Further clarification is also provided in Fig. 1, though it is not clear if this is the actual architecture or just intended as an example. Please (just in a sentence or two) summarize the DNN architecture i.e. clarify precisely the no. of inputs, the number of hidden layers (and nodes in each layer), and the number of outputs to remove any ambiguity. Such details are rather important should one attempt to reproduce this work.

The reviewer hopes that the above points will help to further improve the paper.

---

## Referee Comment (RC4) · Anonymous Referee #3 · 27 Nov 2020

RC

This paper applies a massive forward modeling of tsunami sediment transport and machine learning by deep learning neural network to invert tsunami characteristics, such as inundation distance, flow depth and speed, and input sediment concentration based on sedimentary data of the 2004 Indian Ocean Tsunami collected from the Phra Thong island, Thailand. The authors previously used the inversion model for the deposits of the 2011 Tohoku earthquake tsunami from the Sendai Plain, Japan, and demonstrated the model performance and applicability. Their inversion model is a promising tool to

quantify hydrodynamic parameters of paleotsunamis based on sedimentary deposits, and will benefit assessing hazards from future tsunami inundations and modeling their sources worldwide. Application with different field dataset is necessary to verify, validate and improve the model.

Although the scope is interesting and the achievements are valuable, an extensive improvements will be needed before the publication. One of the major concerns is the technical problems on writing. Sentences in the body text are often complicated and difficult to understand the author's intent. A complete English proofreading by professional services or native speakers is needed. Section 3.2.1, which explains the sedimentary data from the Phra Thong island, must be placed before the description of the inversion results. In addition, earlier papers sometimes were inappropriately cited, and the order of some figures (and insets) are not consistent with the structure of the paper. Therefore, comprehensive reorganization and correction are required to improve the readability of the paper.

Another concern is that whether the model assumption is valid for the study area. For example, both the transect of the tsunami deposit sites and the reference line (Figure 6) is oblique to the coastline, meanwhile the model assumes that the coordinate x for the forward simulation is perpendicular to the shoreline (equation 2). I'm curious that how likely the direction of tsunami inundation was consistent with these lines. Satellite imageries show that the geometry and directions of the sandy ridges are quite complex, implying the tsunami inundation might have been affected the local topography. Fujino et al. (2010) mentioned that the measurement of the flow direction was not many and in fact only single measurement was made near the coastline of the transect. There may be an uncertainty in the tsunami inundation direction (and probably the sediment source). If this is the case, additional computation of the forward and DLNN model using reference line with different directions are needed.

With regard to Section 4.3, I think the flow speed comparison is problematic, since the model assumes a constant flow speed over the inversion region and it is not clear

whether it represents either an average, maximum or something else. This also applies to the measured flow speeds. Unless the attributes of the measured flow speeds (i.e. average, maximum or other) are specified, the measured values cannot be compared with inversion results. The comparison to the inversion results of the TsuSedMod also needs a careful discussion, since the TsuSedMod employs different model assumptions and formulations. It is not clear how the comparison of the two different inversion results are justified.

The idea of coupling DLNN with other tsunami hydrodynamic model, such as the well-validated TUNAMI-N2, is very interesting. Although it must be computationally expensive, the DLNN inversion can include much more physically plausible hydrodynamic models to improve the model performance. I suggest to expand on this aspect, such as outlining a road map and future challenges.

Other minor specific comments are found in the annotated PDF. I hope the authors may find my comments useful for revising the manuscript.

Please also note the supplement to this comment:
https://nhess.copernicus.org/preprints/nhess-2020-373/nhess-2020-373-RC4-supplement.pdf

**Supplement:**

[revised manuscript text omitted]

---

## Author Comment (AC2) · 23 Dec 2020

We thank the reviewer for the critical assessment of our manuscript and for the numerous comments and suggestions. Please find our responses to each comment below (in bold italics).

1. The title does not clearly correspond to the content. Either the authors change the manuscript accordingly and provide clear field validation information or they must remove "tsunami deposits" from the title.

   ***RE: Thank you for your comment. In accordance with your comment, we will add a figure showing the detailed information of the analyzed sediments. We have used the field data from Phra Thong island, Thailand to reconstruct the flow conditions of 2004 Indian Ocean tsunami. We have included our dataset as "Thai_gs5.csv" in https://doi.org/10.5281/zenodo.4075137. The codes include the same dataset to predict the final results. So, we decided to keep the title unchanged.***

2. Are you referring to grain-size? When I look at Figure 9 that is what you are presenting. One thing in sediment concentration and grain-size distribution on the incoming tsunami waves, another totally different is grain-size and packing on the tsunami deposits. Throughout the text is not clear what you are determining, because it was not possible how you estimate sediment concentration on the wave from the deposit if your are solely relying on the inverse model. So, this needs to be clarified and the fact that you did not incorporated any reference to "sediment concentration " on the (minimal) conclusions provide further increases my confusion.

   ***RE: Thank you for the comment. We will add a new diagram to show thickness and grain size distribution from the study area. Here, we refer to the sediment concentration but in Figure 9 we presented volume per unit area and spatial grain-size distribution. We reconstructed sediment concentration of the tsunami deposits.***
   ***In response to your comment we will modify the text in the discussion part and added Goto et al. (2014) reference for clarification in our revised manuscript.***
   ***Our inversion model estimates values of sediment concentration of tsunami that best explain distribution of thickness and grain size of tsunami deposits. The inverse model is trained from the results of the forward model calculation.***

3. Please rewrite. How did you determined the post-tsunami concentration? Was this data based on post-tsunami survey? Please make the text more fluent.

   ***Thank you for the suggestion. We determined the post-tsunami concentration using the DNN inverse model, which automatically finds the distribution of sediment concentration that best***

*realizes the actual observed distribution of thickness and grain size of the tsunami deposit. We will modify the text in our revised manuscript.*
*Revised text: The DNN inverse analysis reconstructed the values of flow conditions such as maximum inundation length, flow velocity and maximum flow depth, sediment concentration of five grain size classes using the datasets of grain-size distribution measured from the post-tsunami survey around Phra Thong island.*

4. Abstract, Agree but the greater challenge is to study older deposits and reconstruct physical parameters from them.

   *RE: We agree with the reviewer.*

5. P-1, Line 15, This sentence is somewhat confusing. Please rewrite.

   **RE: Thank you for the suggestion. We will modify the text in the revised manuscript.**

6. P-2, Line 25, Please add citation.

   *RE: Thank you for identifying this. We have added the citation.*

7. Line 35, Please revise English. The wroding and figures are not well-structured.

   *RE: Thank you for the suggestion. We have revised the text.*

8. Please see Costa et al., 2011, Earth Surface Processes and Landforms, Costa et al., 2012, The Holocene and Moreira et al., 2017, Marine Geology

   *RE: Thank you for the suggestion. We considered adding costa et al., 2011 and Moreira et al., 2017.*

9. P-3, Line 73, The big problem with this manuscript is that is not clear how the inverse model was validated. Figure 9 shows something but much further detail needs to be provided much earlier in the text for the reader to understand what the authors achieved or are aiming.

   *RE: Thank you for the comment. We will provide a paragraph on the workflow of the inverse model for detailed clarifications.*
   *P.3, Line 73, Here, we conduct an DNN inverse analysis of the tsunami deposits measured at Phra Thong island and reconstruct the flow conditions such as the maximum inundation length, flow velocity, maximum flow depth and sediment concentrations of five grain-size classes. The inverse model was based on the forward model, which was proposed by Naruse and Abe, (2017). The forward model calculations were iterated at random initial flow conditions to produce artificial training data sets that represent depositional characteristics such as the spatial distribution of thickness and grain-size composition. Using the artificial training datasets, the DNN was then trained to establish a relation between the depositional characteristics and the*

*and the flow conditions. The post-trained DNN model was ready to predict flow conditions from the tsunami deposits after the performance of the trained DNN was verified using test data sets. The 1-D cubic interpolation was applied to the field data sets of Phra Thong island to fit the dataset to model grids. Finally, this DNN inverse model was applied to the field data sets from the Phra Thong island, Thailand to reconstruct the flow conditions of 2004 Indian Ocean tsunami. We also used the reconstructed flow conditions to estimate the spatial distribution of the volume per unit area and grain-size composition from Phra Thong island and compare the distribution with the measured data. Our inverse model was already validated to be effective for 2011 Tohoku-oki tsunami deposits distributed in Sendai Plain. In case of Phra Thong island, we validated the results by the field measurements of the tsunami flow depth. Also, the estimated thickness and grain size distribution of tsunami deposits were compared with the actual measurements. Our inverse analysis results could be used for designing future tsunami hazard assessments and disaster mitigation strategies in Thailand.*

10. P-3, Line 83, If you mention this, then one would expect to see grain-size variations and deposit thickness data.

    *RE: Thank you for the suggestion. We will add a diagram which represents grain-size distribution and thickness data.*

11. Line 105, This is a big simplification...

    *RE: Thank you for the comment. This simplification was done by Naruse and Abe, 2017 and this was further used by Mitra et al., 2020. For the details of the simplifications and step by step procedures, please refer to Naruse and Abe, 2017.*

12. P-5, Line 115, What one expects from an inverse model (like you are suggesting in the title) is that you depart from field data and use it to reconstruct the physical parameters of the tsunami. This is not what you are explaining here. You use the forward model to produce tsunami hydraulic features and then based on the neural network you check where the data produced is nore robust and assume the values produced. I was expecting a regression from deposits to flow characteristics. It does not seem to be the case here.

    *RE: Thank you for the comment. Yes, we are using the field data to reconstruct the physical parameters using the DNN inverse model. The forward model is used to generate artificial spatial grain size distribution and volume per unit area data of tsunami deposits which was used to train inverse model. We checked the robustness and precision of the inverse model using the artificial test dataset, and then used the produced inverse model to reconstruct the flow conditions of the tsunami from the actual deposit.*

13. P-5, Line 119, I understand it but you need to clearly show where is the data coming from. can you please add a map and grain-size data.

*RE: We assume that this comment is related to comment number 10. Please refer to the reply in comment number 10.*

14. P-5, Line 134, Why 0 to 2% concentration? I do not understand where this value comes from? Is it random? From observations? Where? All the data I have access to suggest (much of it publiched by several authors) much higher sediment concentrations.

    *RE: Thank you for the comment. This range was considered from Goto et al., (2018) sediment concentration analysis which indicates that the total sediment concentration for tsunami deposits is usually around 2%, and therefore the concentration for each grain size class seems unlikely to exceed 2%. In response to your comment we will modify our text by adding this reference. Please refer to the reply of comment 2 where we have written the modified text.*

15. P-5, Line 145, This needs to be explained earlier and more clearly.

    *RE: Thank you for the suggestion. We will revise the text in the revised manuscript.*

16. P-6, Line 157, I understand the seminal character of the work by Mitra et al (2020) and the need to many citations along the manuscript but is some cases I believe this could be avoided.

    *RE: Thank you for the suggestion. We will remove the sentence from the revised manuscript.*

17. P-7, Line 161, What were their charateristics?

    *RE: The characteristics of artificial data sets were depositional characteristics such as volume per unit area and grain-size distribution.*

    *In response to your comment we will revise the text in the revised manuscript.*

18. P-7, Line 165, And between 2000 and 3000 what is the gain?

    *RE: The loss function was optimized already to its lowest value, the calculations continue up to 3000 epochs, the progress will remain the same. There will be no change as it already converged before 2000 epochs.*

19. P-7, Line 169, this is obvious. No need to add it.

    *RE: Thank you for the suggestion. We will remove it from the revised manuscript.*

20. P-8, Line 188, Please add a aerial image with the sampling point clearly marked.

    *RE: We agree with the reviewer. We will add a google earth image with the marked sampling points.*

21. P-8, Line 191, add reference. Heights were reported or modeled?

*RE: Thank you for the suggestion. We will add the reference Fujino et al., (2010) as reference. Heights were reported from the water mark at the study area. We will modify the text by mentioning this in our revised manuscript.*

22. P-8, Line 192, the nearshore area?

    *RE: Yes. The area was nearshore.*

23. P-8, Line 193, thin and finer?

    *RE: The deposits became thinner and finer in the landward direction. We will revise our text by adding both phenomena.*

24. P-8. Line 196, along the analyzed profile, right? Otherwise it contradicts what is stated above regarding maximum thickness of the deposit.

    *RE: Yes, all the thickness mentioned here is along the analyzed profile.*

25. P-8, P 198, This clearly needs to be provided here as well. This is pivotal for the validation and must be displayed in this manuscript as well.

    *RE: We assume that this comment is related to comment number 10. Please refer to the reply in comment number 10.*

26. P-10, Line 200, Not sure if I understood this last sentence correctly. Please provide fruther information. So, you stopped using field data?

    *RE: Thank you for the comment. We have done subsampling test to check the effect of irregularly spaced field datasets on the accuracy of the inversion. The details on the subsampling procedure is given in Mitra et al. (2020).*
    *We agree that we should add some basic information or the add the reference in the text. We will modify the text in our revised manuscript.*

27. P-14, Line 222, Please clarify how you obtained these values.

    *RE: Thank you for the comment. These values are obtained from table 1. We will add clarifications on how we obtained this data in the revised manuscript.*

28. P-14, Line 227, Discussion needs to be rewritten.

    *RE: Thank you for the comment. We hope to modify our discussion accordingly.*

29. P-14P, Line 232, So model was not validated against field data?

*RE: The model reconstructed the tsunami flow characteristics from the actual tsunami deposit, and the predicted results were validated with the field measurements of flow depths. This model was trained with artificial datasets of tsunami deposits and validated with the field data. The present model is estimating fair results using inexpensive artificial data for training of the neural network and avoiding the difficulties to gather large amounts of datasets of tsunami deposits with in-situ measurements of flow velocity and depth. For tsunami deposit dataset in terms of grain size distribution perpendicular to coastline is not easily obtainable and sometimes good datasets are difficult to obtain due to obstructions in field areas, and measurements of flow hydraulic parameters such as velocity were quite rare. Therefore, training the model with the real measurements may not be most viable option in terms of expense and performance. Indeed, all of previous studies on inverse analysis were not developed from the relationship between measurements of deposits and flow parameters because it is practically impossible. Instead, their models were depended on the simplified hydraulic modeling of tsunamis. Even if we can train model with smaller number of training data obtained by measurements in the field, the model tends to overfit which results into poor performance of the model on the observed value.*
*To this end, we employed very different approach, which uses the calculation of results in the forward model as the training datasets.*

30. P-14, Line 239, when you mention true values are you referring to field data?

    *RE: We agree that this true value might create confusion. We referred the true values as the test dataset from artificial data that determines the performance of the model. We will clarify this terminology in the revised manuscript.*

31. P-16, Figure 9, For me this is the key findings of the manuscript and are somewhat lost in the structure. This is a key figure that should be move further up. It will also important to acess raw/original grain-size curves retrieved from the field by Fujino et al. (2010)

    *RE: Thank you for the comment. We considered shifting this diagram in the upper part of the section.*

32. P-17, Line 257, This sub-section is crucial and is poorly explained above and consequently also here. The authors should make an effort to clarify the field validation. I read the manuscript 3 times and struggle to fully understand what field data you used. And if it is an regressive exercise why you do not use thickness and grain-size curve or D50. This needs to be clear for the reader from the start of the manuscript.

    *RE: Thank you for the comment. We understand that there must be lack of clarification in the description of validation regarding the application of the field data. We hope to explain in more details in the revised manuscript.*

33. P-17, Line 266, measured or modeled?

*RE: The flow heights were measured using water mark in the field. We will add this detail in the revised manuscript.*

34. P-17, Line 269, observed or modeled? Did someone measure flow height at these specific points or these values are the result of forward modeling exercises? I suspect it is the latter. So change the text, please to be accurate.

*RE: Yes, the mentioned groups measured the flow heights at specific locations. The details are available in the link, http://www.nda.ac.jp/ fujima/TMD/fujicom.html.*
*In response to your comment we will unify the terminologies of measured flow heights.*

35. P-17, Line 274, But you used Fujino et al., 2010, right?

*RE: We applied the dataset of volume per unit area and grain size distribution in our model and the model results were validated with several reported or measured values by different researchers. Here, the distance 400 m indicates the locations of the measured value by Jankaew et al. (2011, 2008) from our study area.*

36. P-18, Line 296, But there is a wide range of values presented even in this manuscript introduction and study area?!?!

*RE: The DNN inverse model estimates only a single value. We have added the uncertainty or error estimations. We used a range of artificial values to generate artificial dataset of depositional characteristics in the forward model. Using that artificial data sets the inverse model was trained and the final predicted result was a single value for one sampling window size. For example, we used 1700 sampling window size to reconstruct the single values for each parameter of flow conditions.*

37. P-18, Line, 307, Please explain this idea better and in greater detail.

*RE: Thank you for your comment. We will modify the text and some further details in the last paragraph.*

38. P-18, Line 310, Where have you provided this information? Where is the detailed variation of tsunami deposit thickness variation in the field and its comparison with the model results?

*RE: Figure 9 shows the tsunami deposit volume per unit area and its comparison with the model results.*

39. P-18, Line 315, Please rewrite the Conclusions.

*RE: Thank you for the suggestion. We agree that we should rewrite the conclusion in more details. We will do so in our revised manuscript.*

---

## Author Comment (AC3) · 23 Dec 2020

We thank the reviewer for the insightful assessment of our manuscript and for the numerous comments and suggestions. We have provided answers to your questions as listed below (in bold italics).

Q1: The figures presented as Figure 1 in this paper have already appeared in M2020 (their Figs. 1 and 2). This is at least acknowledged in the caption. Figure 2 also appears identical to Fig. 3 of M2020 (apart from some color changes), but this does not seem to be indicated in the Figure caption. Is this indeed essentially the identical figure? If so, this ought to be acknowledged. (It could just be very similar, and the differences not perceptable).

***RE: Thank you for identifying this. We agree that Figure 1 in this manuscript looks similar to the appeared figure in Mitra et al., (2020). However, we have made slight changes in the Figure 1 by changing the number of grain size classes from six o five, hence the number of output nodes should be 8, here we did the typo by mentioning the number of nodes as 9. We will correct the typo and add the reference of Mitra et al., (2020). For Figure 2, although the figure looks similar and imperceptible, we changed the number of training datasets and the performance of loss function is slightly different from the previous paper as the grain size distribution is different for 2004 Indian ocean tsunami at Phra Thong island. Thus, we generated separate artificial dataset and that Figure 2 shows that performance.***

Q2: Several details of the numerical model used to generate test data sets in Section 2.1 are seemingly missing. These include, the following issues: How are the friction velocity ($u\_*$), the setting velocity ($w\_{s,i}$), the sediment entrainment coefficient ($E\_{si}$), and other variables ($r\_{0i}$ and $F\_i$) determined?

***RE: Thank you for your suggestion. We did not add the details of the parameters mentioned as we thought it would be repetitive as we the details are already in Naruse and Abe, 2017 and Mitra et al. 2020. In response to your comment we will either add the details of the parameters or cite the reference properly in our revised manuscript.***

***Q3:*** Regarding the friction velocity ($u\_*$), in particular, it should be noted that great care ought to be taken for this quantity if standard (based on steady flow) friction formulas are used, as several recent research papers have shown that tsunami induced bounday layers may span only a fraction of the water depth, and hence these may well be invalid. See e.g. Lacy et al. (DOI: 10.1029/2012JC007954), Williams & Fuhrman (DOI: 10.1016/j.coastaleng.2015.12.002), Tinh & Tanaka (DOI: 10.1080/21664250.2019.1672127) or Larsen & Fuhrman (DOI: 10.1016/j.coastaleng.2019.04.011). Please clarify this point, and if this is indeed being done, this potential defficiency ought to at least be acknowledged.

*RE: We understand the point the reviewer makes. We have used standard friction formula for our model. In response to your comment, we will add the mentioned references and will acknowledge this potential deficiency in our revised manuscript.*

Q3: Can a definition sketch of the model domain (etc.) being used for the generation of the training data sets please be provided? This will help readers immensely to get an idea as to the actual setup being used. Plots just showing performance (like Figs. 2 and 3) fail to provide this.

*RE: Thank you for the comment. We will add a diagram on the explanation of model configuration in the revised manuscript.*

Q4: I do not find that the DNN architecture being used is presented with sufficient clarity. In Section 2.2 (top) it is stated that the DNN model accepts grain-size and thickness distribution at an input layer, and that the outputs are the "tsunami characteristics through several hidden layers". This is rather unclear. Further clarification is also provided in Fig. 1, though it is not clear if this is the actual architecture or just intended as an example. Please (just in a sentence or two) summarize the DNN architecture i.e. clarify precisely the no. of inputs, the number of hidden layers (and nodes in each layer), and the number of outputs to remove any ambiguity. Such details are rather important should one attempt to reproduce this work.

*RE: Thank you for the valuable suggestion. We agree with the reviewer that we should add the details of input and output layers. We will add one paragraph in methodology section, 2.2 on the workflow inside the DNN architecture and will mention about the number of inputs, output nodes, hidden layers.*

---

## Author Comment (AC5) · 5 Jan 2021

We thank the reviewer for the insightful assessment of our manuscript and for the numerous comments and suggestions. We have already provided the logical reasoning to the comments that reviewers suggested. However, while analyzing the detailed natural dataset repetitively, we found that there was an error in the representative diameters for grain size classes, but the weight percentage values are correct for the dataset of Phra Thong island. The error was there only with the values of the representative diameters. The grain size classes which were mentioned in

the submitted manuscript were 659, 329, 164, 82 and 41 $\mu$m respectively which will be changed to 726, 364, 182, 91 and 46 $\mu$m respectively in the revised manuscript. Hence, the DNN inversion results will be slightly changed in values. We have checked the DNN results with the revised dataset and the results were almost similar with the reported values in the manuscript. The final changes will be done on the revised manuscript. I will also upload the revised and detailed calculated dataset to zenodo repository (https://doi.org/10.5281/zenodo.4075137). We apologize for the inconvenience caused. I have added the revised "Thai_gs5_revised.csv" data as a supplement with this comment.

Please also note the supplement to this comment:
https://nhess.copernicus.org/preprints/nhess-2020-373/nhess-2020-373-AC5-supplement.pdf

---

## Author Comment (AC6) · 5 Jan 2021

We thank the reviewer for the insightful assessment of our manuscript and for the numerous comments and suggestions. We have already provided the logical reasoning to the comments that reviewers suggested. However, while analyzing the detailed natural dataset repetitively, we found that there was an error in the representative diameters for grain size classes, but the weight percentage values are correct for the dataset of Phra Thong island. The error was there only with the values of the representative diameters. The grain size classes which were mentioned in

the submitted manuscript were 659, 329, 164, 82 and 41  $\mu$ m respectively which will be changed to 726, 364, 182, 91 and 46  $\mu$ m respectively in the revised manuscript. Hence, the DNN inversion results will be slightly changed in values. We have checked the DNN results with the revised dataset and the results were almost similar with the reported values in the manuscript. The final changes will be done on the revised manuscript. I will also upload the revised and detailed calculated dataset to zenodo repository (https://doi.org/10.5281/zenodo.4075137.). We apologize for the inconvenience caused. I have added the revised dataset as a supplement with this comment.

Please also note the supplement to this comment: https://nhess.copernicus.org/preprints/nhess-2020-373/nhess-2020-373-AC6supplement.pdf

**Supplement:**

| distance | 726 | 364 | 182 | 91 | 46 |
|---|---|---|---|---|---|
| 0 | 0.010399 | 0.029123 | 0.021142 | 0.006882 | 0.000455 |
| 51.41541 | 0.004638 | 0.010623 | 0.016518 | 0.01624 | 0.001981 |
| 101.5412 | 0.01909 | 0.033383 | 0.021191 | 0.015271 | 0.002038 |
| 132.8777 | 0.014846 | 0.036879 | 0.038916 | 0.028934 | 0.00338 |
| 178.3496 | 0.000897 | 0.004421 | 0.032821 | 0.048209 | 0.008753 |
| 235.2521 | 0.000248 | 0.00071 | 0.00996 | 0.015168 | 0.001914 |
| 267.1548 | 0.023041 | 0.013593 | 0.010173 | 0.017016 | 0.003069 |
| 296.4631 | 0.012222 | 0.011268 | 0.008694 | 0.012747 | 0.002035 |
| 344.9381 | 0.004596 | 0.009952 | 0.017344 | 0.020869 | 0.003623 |
| 414.1045 | 0.002789 | 0.004262 | 0.019079 | 0.031537 | 0.006802 |
| 458.8842 | 0.00045 | 0.000735 | 0.006082 | 0.009064 | 0.001527 |
| 491.7154 | 0.006784 | 0.018716 | 0.026314 | 0.023898 | 0.003289 |
| 525.9441 | 0.000312 | 0.00067 | 0.008444 | 0.013288 | 0.002287 |
| 607.6578 | 0.006453 | 0.015906 | 0.031496 | 0.038024 | 0.007074 |
| 673.2236 | 0.004498 | 0.011375 | 0.017926 | 0.019306 | 0.003352 |
| 751.9185 | 0.00239 | 0.005526 | 0.013929 | 0.014474 | 0.002017 |
| 807.2173 | 0.001888 | 0.00435 | 0.016767 | 0.024377 | 0.005193 |
| 868.3649 | 0.001695 | 0.005897 | 0.015554 | 0.017724 | 0.002917 |
| 956.0379 | 0.00281 | 0.009894 | 0.029122 | 0.034147 | 0.005493 |
| 1017.972 | 0.003594 | 0.009873 | 0.026584 | 0.032365 | 0.005828 |
| 1091.84 | 0.001516 | 0.003771 | 0.016798 | 0.024389 | 0.005023 |
| 1175.299 | 0.001224 | 0.001481 | 0.009721 | 0.01953 | 0.005333 |
| 1253.758 | 0.001143 | 0.001684 | 0.008231 | 0.014364 | 0.00333 |
| 1324.607 | 0.00114 | 0.001083 | 0.006477 | 0.013431 | 0.003911 |
| 1413.868 | 0.000597 | 0.001263 | 0.005764 | 0.009499 | 0.002394 |
| 1489.356 | 0.000723 | 0.001125 | 0.004054 | 0.008971 | 0.002817 |
| 1537.138 | 0.000691 | 0.000728 | 0.003361 | 0.006853 | 0.002003 |
| 1641.806 | 0.001587 | 0.001434 | 0.003044 | 0.005171 | 0.001253 |
| 1782.245 | 0.002549 | 0.004872 | 0.001523 | 0.001276 | 0.000657 |

---

## Author Response (AR1)

**et al.,**
We thank the reviewer for the critical assessment of our manuscript and for the numerous comments
and suggestions. We have provided answers to your questions as listed below (in bold italics).

Q1: The Science is there but English must be revised extensively and above all there must be a clear
clarification on how and what exactly field data was used to validate the model. I believe you used
Fujino et al. (2010) data but sometimes when reading the manuscript, one feels puzzled to confirm
that you did use it and which values have you used it. For instance, thickness, grain-size curve, etc.
Therefore, I cannot agree with the title proposed because it was not well-establish that the regressive
model used geological data. Sometimes the reader feels, the models fed and validated each other,
and no solid, extensive and accurate field data was used. I assumed that this might just be a language
and writing problem. Even if it is that, you need to address it. Sometimes the text is confusing and
one wonders what you trying to transmit. For example, when you state the model was validated by
"observed" flow depths in several locations along the studied profile...in fact, you are saying that the
model agrees well with previous modelling exercises for flow depth establishment. The meaning of
both sentences in totally different regarding field validation and this is crucial for this manuscript.
*RE: The authors would like to thank Dr. Costa for his comments. We made a substantial effort*
*on the clarifications and the overall organization of the paper. We have done the English*
*language and grammar checking on our manuscript by an English proof-reading service agency*
*for journals.*
*Yes, we have used the data set from Fujino et al. (2010) and the data set is given as*
*"Thai_gs5_revised_1.csv" in https://doi.org/10.5281/zenodo.4511317. However, we have revised*
*the representative diameters for grain-size classes in our revised manuscript and the detailed*
*calculation is provided in the above-mentioned repository.*
*We added an additional diagram as Figure 2 and 3 on page no. 5 and 6, on the distribution of the*
*thickness of the samples with the distance along with the mean grain size and segregation of the*
*grain size classes from the distribution. Hence, we decided to keep the same title as it uses the real*
*field data set in the inverse model and the model uses mean squared error for the regression*
*algorithm.*
*Here, "observed" flow depth implies the measured flow depth. We have unified the terminologies*
*for all measured values to avoid further confusions.*
Q2: There are many other aspects I raised on the annotated version and I suggest you analyse them
critically. I might have misunderstood some wording (which means that you need to make it clear)
or I might have perceived things correctly (which means you need to change the structure and scope
of the manuscript). One example, is sediment concentration. How can you validate flow sediment
concentration from the deposit? Only if you look at grain-size curve, spatial distribution and packing
(inner architectural arrangement) of the deposit. You never mention this along the manuscript which
means that I am puzzled how you reconstruct sediment concentration on the inverse model. It is easy to understand how you do it with the forward model but departing from sediments (without
mentioning the characteristics above) is baffling.
*RE: Thank you for your feedback. Regarding sediment concentration, we did not intend to*
*validate the sediment concentration as it is almost impossible to evaluate the reconstructed values*
*of sediment concentration because there are no available observational data. We only*
*reconstructed the values of sediment concentration using DNN inverse model. However, Goto et*
*al. (2014) used the entire thickness and measured inundation depth to estimate the sediment*
*concentration which was around 2% in the inundation flow of the 2011 Tohoku-oki tsunami.*
*Hence, we added the reference of sediment concentration on Page 10., line 205.*
*We have provided the explanation of all comments in reply of annotated document.*
**Minor comments (annotations):**
**Interactive comment on "Reconstruction of flow conditions from 2004 Indian Ocean tsunami**
**deposits at the Phra Thong island using a deep neural network inverse model" by Rimali Mitra**
**et al.,**
We thank the reviewer for the critical assessment of our manuscript and for the numerous comments
and suggestions. Please find our responses to each comment below (in bold italics).
1. The title does not clearly correspond to the content. Either the authors change the manuscript
accordingly and provide clear field validation information or they must remove "tsunami deposits"
from the title.
*RE: Thank you for your comment. In accordance with your comment, we have added figure 3 to*
*show the detailed information of the analyzed sediments. We have used the field data from Phra*
*Thong island, Thailand to reconstruct the flow conditions of 2004 Indian Ocean tsunami. Yes, we*
*have used the data set from Fujino et al. (2010) and the data set is given as*
*"Thai_gs5_revised_1.csv" in https://doi.org/10.5281/zenodo.4511317. However, we have revised*
*the representative diameters for grain-size classes in our revised manuscript and the detailed*
*calculation is provided in the above-mentioned repository. The codes include the same data set to*
*predict the results. So, we decided to keep the title unchanged.*
2. Are you referring to grain-size? When I look at Figure 9 that is what you are presenting. One thing
in sediment concentration and grain-size distribution on the incoming tsunami waves, another
totally different is grain-size and packing on the tsunami deposits. Throughout the text is not clear
what you are determining, because it was not possible how you estimate sediment concentration on
the wave from the deposit if your are solely relying on the inverse model. So, this needs to be
clarified and the fact that you did not incorporated any reference to "sediment concentration " on
the (minimal) conclusions provide further increases my confusion.
*RE: Thank you for the comment.*
*P.5, 6, we added two new diagrams as Figure 2 and 3 to show thickness and grain size distribution*
*from the study area. Here, we refer to the sediment concentration but in Figure 9 we presented*

*volume per unit area and spatial grain-size distribution. We reconstructed sediment concentration*
*of the tsunami deposits.*
*Our inversion model estimates values of sediment concentration of tsunami that best explain*
*distribution of thickness and grain size of tsunami deposits. The inverse model is trained from the*
*results of the forward model calculation.*
*In response to your comment have added the reference Goto et al. (2014) reference and*
*clarification in our revised manuscript as follows,*
*P. 10, line 202, The range of parameters adopted in this study is applicable to most of the large-*
*scale tsunami-inundated areas as the ranges have been selected with several case studies of*
*tsunamis that includes mostly field measurements, survivor video and numerical analysis (Mori et*
*al., 2011; Wijetunge, 2006; Szczucínski et al., 2012; Matsutomi and Okamoto,2010; Abe et al.,*
*2012; Fritz et al., 2006; Nandasena et al., 2012; Goto et al., 2014).*
3. Please rewrite. How did you determined the post-tsunami concentration? Was this data based on
post-tsunami survey? Please make the text more fluent.
*Thank you for the suggestion. We determined the post-tsunami concentration using the DNN*
*inverse model, which automatically finds the distribution of sediment concentration that best*
*realizes the actual observed distribution of thickness and grain size of the tsunami deposit. We will*
*modify the text in our revised manuscript.*
*Revised text: P.1, line 5, The DNN inverse analysis reconstructed the values of flow conditions*
*such as maximum inundation distance, flow velocity and maximum flow depth, sediment*
*concentration of five grain-size classes using the thickness and grain-size distribution of the*
*tsunami deposit from the post-tsunami survey around Phra Thong island.*
4.  Abstract, Agree but the greater challenge is to study older deposits and reconstruct physical
parameters from them.
*RE: We agree with the reviewer.*
5.  P-1, Line 15, This sentence is somewhat confusing. Please rewrite.
*RE: Thank you for the suggestion. We modified the text in the revised manuscript as follows:*
*P.1, line 18, The total damage was estimated to amount to around USD 508 million, which equates*
*to 2.2% of GDP while the number of deaths was 4225, with the injured and missing cases.*
6.  P-2, Line 25, Please add citation.
*RE: Thank you for identifying this. We have added the citation Suppasri et al., 2015 .*
7.  Line 35, Please revise English. The wroding and figures are not well-structured.
*RE: Thank you for the suggestion. We have revised the text.*

8. Please see Costa et al., 2011, Earth Surface Processes and Landforms, Costa et al., 2012, The
Holocene and Moreira et al., 2017, Marine Geology

*RE: Thank you for the suggestion. We considered adding costa et al., 2011 and Moreira et al.,*
*2017.*

1399. P-3, Line 73, The big problem with this manuscript is that is not clear how the inverse model was
validated. Figure 9 shows something but much further detail needs to be provided much earlier in
the text for the reader to understand what the authors achieved or are aiming.

*RE: Thank you for the comment. We have provided a paragraph on the workflow of the inverse*
*model for detailed clarifications.*
*P.3, Line 86, Here, we conduct an DNN inverse analysis of the tsunami deposits measured at Phra*
*Thong island and reconstruct theflow conditions such as the maximum inundation length, flow*
*velocity, maximum flow depth and sediment concentrations offive grain-size classes. The inverse*
*model was based on the forward model, which was proposed by Naruse and Abe (2017).The*
*forward model calculations were iterated at random initial flow conditions to produce artificial*
*training data sets thatrepresent depositional characteristics such as the spatial distribution of*
*thickness and grain-size composition. Using the artificial903*
*training data sets, the DNN was then trained to establish a relation between the depositional*
*characteristics and the and the flowconditions. The post-trained DNN model was ready to predict*
*flow conditions from the tsunami deposits after the performanceof the trained DNN was verified*
*using test data sets. The 1-D cubic interpolation was applied to the field data sets of PhraThong*
*island to fit the data set to model grids. Finally, this DNN inverse model was applied to the field*
*data sets from thePhra Thong island, Thailand to reconstruct the flow conditions of 2004 Indian*
*Ocean tsunami. Our inverse model was already95validated to be effective for 2011 Tohoku-oki*
*tsunami deposits distributed in Sendai Plain (Mitra et al., 2020). In case of PhraThong island, we*
*validated the results by the field measurements of the tsunami flow depth. Also, the estimated*
*thickness andgrain size distribution of tsunami deposits were compared with the actual*
*measurements. Our inverse analysis results couldbe used for designing future tsunami hazard*
*assessments and disaster mitigation strategies in Thailand.*

10. P-3, Line 83, If you mention this, then one would expect to see grain-size variations and deposit
thickness data.

*RE: Thank you for the suggestion. We have added Figure 2 and 3 which represents grain-size*
*distribution and thickness data.*

17111. Line 105, This is a big simplification...

*RE: Thank you for the comment. This simplification was done by Naruse and Abe, 2017 and this*
*was further used by Mitra et al., 2020. For the details of the simplifications and step by step*
*procedures, please refer to Naruse and Abe, 2017.*

12. P-5, Line 115, What one expects from an inverse model (like you are suggesting in the title) is that
you depart from field data and use it to reconstruct the physical parameters of the tsunami. This is
not what you are explaining here. You use the forward model to produce tsunami hydraulic features
and then based on the neural network you check where the data produced is nore robust and assume
the values produced. I was expecting a regression from deposits to flow characteristics. It does not
seem to be the case here.

*RE: Thank you for the comment. Yes, we are using the field data to reconstruct the physical*
*parameters using the DNN inverse model. The forward model is used to generate artificial spatial*
*grain size distribution and volume per unit area data of tsunami deposits which was used to train*
*inverse model. We checked the robustness and precision of the inverse model using the artificial*
*test data set, and then used the produced inverse model to reconstruct the flow conditions of the*
*tsunami from the actual deposit.*

13. P-5, Line 119, I understand it but you need to clearly show where is the data coming from. can you
please add a map and grain-size data.

*RE: We assume that this comment is related to comment number 10. Please refer to the reply in*
*comment number 10.*

14. P-5, Line 134, Why 0 to 2% concentration? I do not understand where this value comes from? Is it
random? From observations? Where? All the data I have access to suggest (much of it publiched
by several authors) much higher sediment concentrations.

*RE: Thank you for the comment. This range was considered from Goto et al., (2018) sediment*
*concentration analysis which indicates that the total sediment concentration for tsunami deposits*
*is usually around 2%, and therefore the concentration for each grain size class seems unlikely to*
*exceed 2%. In response to your comment have modified our text by adding this reference. Please*
*refer to the reply of comment 2 where we have written the modified text.*

15. P-5, Line 145, This needs to be explained earlier and more clearly.

*RE: Thank you for the suggestion. We have revised the text in the revised manuscript.*
*P.11, line 215, To apply the inverse model to the measured values of field data set from Phra Thong*
*island in 1-D vectors, the collected datapoints must be fit into that fixed coordinate system of the*
*model.*

16. P-6, Line 157, I understand the seminal character of the work by Mitra et al (2020) and the need to
many citations along the manuscript but is some cases I believe this could be avoided.

*RE: Thank you for the suggestion. We have removed the sentence from the revised manuscript.*

17. P-7, Line 161, What were their charateristics?

*RE: The characteristics of artificial data sets were depositional characteristics such as volume per unit area and grain-size distribution.*

*In response to your comment we have revised the text in the revised manuscript.*
*P. 11, line 231, The DNN was trained using artificial data sets which were the depositional characteristics such as volume per unit area and grain-size distribution.*

18. P-7, Line 165, And between 2000 and 3000 what is the gain?

*RE: The loss function was optimized already to its lowest value, the calculations continue up to 3000 epochs, the progress will remain the same. There will be no change as it already converged before 2000 epochs.*

19. P-7, Line 169, this is obvious. No need to add it.

*RE: Thank you for the suggestion. We have removed the sentence from the revised manuscript.*

20. P-8, Line 188, Please add a aerial image with the sampling point clearly marked.

*RE: We agree with the reviewer. We have added a google earth image with the marked sampling points (Figure 2).*

21. P-8, Line 191, add reference. Heights were reported or modeled?

*RE: Thank you for the suggestion. We have added the references Jankaew et al., 2008, Fujino et al., 2010. Heights were reported from the above-mentioned literatures.*

22. P-8, Line 192, the nearshore area?

*RE: Yes. The area was nearshore.*

23. P-8, Line 193, thin and finer?

*RE: The deposits became thinner and finer in the landward direction. We have revised the text by adding both phenomena in P. 4, line 122.*

24. P-8. Line 196, along the analyzed profile, right? Otherwise it contradicts what is stated above regarding maximum thickness of the deposit.

*RE: Yes, all the thickness mentioned here is along the analyzed profile.*

25. P-8, P 198, This clearly needs to be provided here as well. This is pivotal for the validation and must be displayed in this manuscript as well.

*RE: We assume that this comment is related to comment number 10. Please refer to the reply in comment number 10.*

26. P-10, Line 200, Not sure if I understood this last sentence correctly. Please provide fruther information. So, you stopped using field data?

*RE: Thank you for the comment. We have done subsampling test to check the effect of irregularly spaced field data sets on the accuracy of the inversion. The details on the subsampling procedure is given in Mitra et al. (2020). In response to your comment, we have modified the text as follows, P. 15, line 262, This test was done to check the effect of irregularly spaced field data sets on the accuracy of the inversion. The details on the subsampling procedure is given in Mitra et al. (2020)*

27. P-14, Line 222, Please clarify how you obtained these values.

*RE: Thank you for the comment. These values are obtained from table 1.  We have added the reference of table 1 in P. 15, line 275.*

28.  P-14, Line 227, Discussion needs to be rewritten.

*RE: Thank you for the comment. We have modified the overall discussion accordingly.*

29. P-14, Line 232, So model was not validated against field data?

*RE: The model reconstructed the tsunami flow characteristics from the actual tsunami deposit, and the predicted results were validated with the field measurements of flow depths. This model was trained with artificial data sets of tsunami deposits and validated with the field data. The present model is estimating fair results using inexpensive artificial data for training of the neural network and avoiding the difficulties to gather large amounts of data sets of tsunami deposits with in-situ measurements of flow velocity and depth. For tsunami deposit data set in terms of grain size distribution perpendicular to coastline is not easily obtainable and sometimes good data sets are difficult to obtain due to obstructions in field areas, and measurements of flow hydraulic parameters such as velocity were quite rare. Therefore, training the model with the real measurements may not be most viable option in terms of expense and performance. Indeed, all of previous studies on inverse analysis were not developed from the relationship between measurements of deposits and flow parameters because it is practically impossible. Instead, their models were depended on the simplified hydraulic modeling of tsunamis. Even if we can train model with smaller number of training data obtained by measurements in the field, the model tends to overfit which results into poor performance of the model on the observed value.*
*To this end, we employed very different approach, which uses the calculation of results in the forward model as the training data sets.*

30.  P-14, Line 239, when you mention true values are you referring to field data?

*RE: We agree that this true value might create confusion. We referred the true values as the test data set from artificial data that determines the performance of the model.*

*P. 20, line 295, Regarding the deviation of the predicted values from the true values which are artificial test data set, the sample standard deviation values were relatively small for all parameters.*

31. P-16, Figure 9, For me this is the key findings of the manuscript and are somewhat lost in the structure. This is a key figure that should be move further up. It will also important to acess raw/original grain-size curves retrieved from the field by Fujino et al. (2010)

*RE: Thank you for the comment. We considered shifting this diagram in the upper part of the section.*

32. P-17, Line 257, This sub-section is crucial and is poorly explained above and consequently also here. The authors should make an effort to clarify the field validation. I read the manuscript 3 times and struggle to fully understand what field data you used. And if it is an regressive exercise why you do not use thickness and grain-size curve or D50. This needs to be clear for the reader from the start of the manuscript.

*RE: Thank you for the comment. We understand that there must be lack of clarification in the description of validation regarding the application of the field data. We have modified the text as follows:*

*P. 21, line 335, The maximum and measured flow heights from Phra Thong island were reported 7.1 m and 5.5 m respectively (\url{http://www.nda.ac.jp/~fujima/TMD/fujicom.html}). The corresponding maximum and minimum values of elevation are 3.1 and 1.1 m respectively (Jankaew et al., 2008, 2011; Brill et al., 2012b). Hence, the approximate estimate of measured maximum flow depth is ranged from 2.4 m to 6.0 m. Considering the bias correction of 0.43 m, the reconstructed value of maximum flow depth (5.3 m) falls within the range of measured maximum flow depth values.*

33. P-17, Line 266, measured or modeled?

*RE: The flow heights were measured using water mark in the field.*

34. P-17, Line 269, observed or modeled? Did someone measure flow height at these specific points or these values are the result of forward modeling exercises? I suspect it is the latter. So change the text, please to be accurate.

*RE: Yes, the mentioned groups measured the flow heights at specific locations. The details are available in the link, http://www.nda.ac.jp/ fujima/TMD/fujicom.html.*

*In response to your comment we have unified the terminologies of measured flow heights.*

35. P-17, Line 274, But you used Fujino et al., 2010, right?

*RE: We applied the data set of volume per unit area and grain size distribution in our model and the model results were validated with several reported or measured values by different researchers.*

*Here, the distance 400 m indicates the locations of the measured value by Jankaew et al. (2011, 2008) from our study area.*

36. P-18, Line 296, But there is a wide range of values presented even in this manuscript introduction and study area?!?!

*RE: The DNN inverse model estimates only a single value. We have added the uncertainty or error estimations. We used a range of artificial values to generate artificial data set of depositional characteristics in the forward model. Using that artificial data sets the inverse model was trained and the final predicted result was a single value for one sampling window size. For example, we used 1700 sampling window size to reconstruct the single values for each parameter of flow conditions.*

37. P-18, Line, 307, Please explain this idea better and in greater detail.

*RE: Thank you for your comment. We have modified the text as follows:*

*P. 22, line 370, In addition to the source model, this model also includes tsunami sediment transport calculation that consists of bed load layer and suspended load layer. However, the calculated value of the sediment thickness was overestimated as the assumption of movable bed for a large area caused excessive erosion of the ground (Masaya et al., 2019).*
*Line 379, The model calculation of Masaya et al. (2019) relies on the estimation of a385single set of fault parameters, which were not widely explored to obtain the optimal parameters. In future, Model TUNAMI-N2can be potentially used as the forward model in DNN inverse model to consider two-dimensional behavior of tsunamis. To do so, the model needs to be modified for considering sediment transport of multiple grain size classes.*

38. P-18, Line 310, Where have you provided this information? Where is the detailed variation of tsunami deposit thickness variation in the field and its comparison with the model results?

*RE: Figure 9 shows the tsunami deposit volume per unit area and its comparison with the model results.*

39. P-18, Line 315, Please rewrite the Conclusions.

*RE: Thank you for the suggestion. We have revised the conclusion as follows:*
*P. 23, line 386, The value of maximum flow depth including the additional bias correction was 5.3 m that was within the range 2.4 m to 6.0 m which was the approximate estimate of measured maximum flow depth at Phra Thong island. The value of flow velocity was also close to the reported values using the video footage from the vicinity of the Phra Thong island. The uncertainty of the results using jackknife method also indicated that simulated results did not contain a large range of values. Phra thong island was one of the most well preserved and historically important area for paleotsunami deposits. Hence, the application of the DNN inverse model was suitable to*

*reconstruct flow conditions of 2004 Indian Ocean tsunami from Phra thong island. The DNN inverse model also represented the comparison of the calculated and measured spatial distribution of volume per unit area along the transect at the island. This model can be applied to any areas of modern and ancient tsunami deposits consisting of low land or flat areas to successfully reconstruct the tsunami flow conditions and can serve as a tool for tsunami hazard assessment and disaster resilience at coastal cities.*

**Reply to Anonymous Referee 2, (NHESS)**
We thank the reviewer for the insightful assessment of our manuscript and for the numerous comments and suggestions. We have provided answers to your questions as listed below (in bold italics).

Q1: The figures presented as Figure 1 in this paper have already appeared in M2020 (their Figs. 1 and 2). This is at least acknowledged in the caption. Figure 2 also appears identical to Fig. 3 of M2020 (apart from some color changes), but this does not seem to be indicated in the Figure caption. Is this indeed essentially the identical figure? If so, this ought to be acknowledged. (It could just be very similar, and the differences not perceptible).

*RE: Thank you for identifying this. We agree that Figure 1 in this manuscript looks similar to the appeared figure in Mitra et al., (2020). However, we have made slight changes in the Figure 1 by changing the number of grain size classes from six o five, hence the number of output nodes should be 8, here we did the typo by mentioning the number of nodes as 9. We have corrected the typo and added the reference of Mitra et al., (2020). In the revised manuscript, although the figure 4 looks similar and imperceptible, we changed the number of training datasets and the performance of loss function is slightly different from the previous paper as the grain size distribution is different for 2004 Indian ocean tsunami at Phra Thong island. Thus, we generated separate artificial dataset and that Figure 4 shows that performance.*

Q2: Several details of the numerical model used to generate test data sets in Section 2.1 are seemingly missing. These include, the following issues: How are the friction velocity ($u\_*$), the setting velocity ($w\_s,i$), the sediment entrainment coefficient ($E\_si$), and other variables ($r\_0i$ and $F\_i$) determined?

*RE: Thank you for your suggestion. We did not add the details of the parameters mentioned as we thought it would be repetitive as we the details are already in Naruse and Abe, 2017 and Mitra et al. 2020. In response to your comment we have added the reference in our revised manuscript.*

*P.7, line 154, The details of the parameters and variables are provided in Naruse and Abe (2017).*

*Q3:* Regarding the friction velocity (u_*), in particular, it should be noted that great care ought to be taken for this quantity if standard (based on steady flow) friction formulas are used, as several recent research papers have shown that tsunami induced bounday layers may span only a fraction of the water depth, and hence these may well be invalid. See e.g. Lacy et al. (DOI: 10.1029/2012JC007954), Williams & Fuhrman (DOI: 10.1016/j.coastaleng.2015.12.002), Tinh & Tanaka (DOI: 10.1080/21664250.2019.1672127) or Larsen & Fuhrman (DOI: 10.1016/j.coastaleng.2019.04.011). Please clarify this point, and if this is indeed being done, this potential defficiency ought to at least be acknowledged.

*RE: We understand the point the reviewer makes. We have used standard friction formula for our model. In response to your comment, we have added the suggested references and revised the text as follows:*

*P. 7, line 145, Here, we employed the flow resistance law to obtain friction velocity using the friction coefficient, which is widely used in general. A few researchers recently reported that tsunami induced boundary layers may span only a fraction of water length formula (Williams and Fuhrman, 2016; Lacy et al., 2012; Larsen and Fuhrman, 2019). The importance of the resistance law for the inverse analysis, considering such non-steady conditions, maybe a subject for future study.*

Q3: Can a definition sketch of the model domain (etc.) being used for the generation of the training data sets please be provided? This will help readers immensely to get an idea as to the actual setup being used. Plots just showing performance (like Figs. 2 and 3) fail to provide this.

*RE: Thank you for the comment. We have added a diagram (figure 5) on the explanation of model configuration in the revised manuscript.*

Q4: I do not find that the DNN architecture being used is presented with sufficient clarity. In Section 2.2 (top) it is stated that the DNN model accepts grain-size and thickness distribution at an input layer, and that the outputs are the "tsunami characteristics through several hidden layers". This is rather unclear. Further clarification is also provided in Fig. 1, though it is not clear if this is the actual architecture or just intended as an example. Please (just in a sentence or two) summarize the DNN architecture i.e. clarify precisely the no. of inputs, the number of hidden layers (and nodes in each layer), and the number of outputs to remove any ambiguity. Such details are rather important should one attempt to reproduce this work.

*RE: Thank you for the valuable suggestion. We agree with the reviewer that we should add the details of input and output layers. We have added the details of the DNN architecture in the revised manuscript. The details are as follows:*

*P. 8, line 179, The DNN structure includes the input layer which consists of input nodes where the input values are the volume per unit area of each grain-size class at the spatial grids. Thus, expression of the input nodes numbers is presented as M×N where M and N are the total number of spatial grids and grain-size classes, respectively. In this inverse model, the total numbers of layers were five among which, the number of hidden layers were three with the 2500 nodes (Mitra*

*et al., 2020). Finally, the output layer consists of the predicted parameters of flow conditions. The*
*details of hyperparameters selection is provided in Mitra et al. (2020).*
**Reply to Anonymous Referee 3, (NHESS)**

**Interactive comment on "Reconstruction of flow conditions from 2004 Indian Ocean tsunami**
**deposits at the Phra Thong island using a deep neural network inverse model" by Rimali Mitra**
**et al.,**

We thank the reviewer for the insightful assessment of our manuscript and for the numerous
comments and suggestions. We have provided answers to your questions as listed below (in bold
italics).

**Major comments:**

Q1: One of the major concerns is the technical problems on writing. Sentences in the body text are
often complicated and difficult to understand the author's intent. A complete English proofreading
by professional services or native speakers is needed. Section 3.2.1, which explains the sedimentary
data from the Phra Thong island, must be placed before the description of the inversion results. In
addition, earlier papers sometimes were inappropriately cited, and the order of some figures (and
insets) are not consistent with the structure of the paper. Therefore, comprehensive reorganization
and correction are required to improve the readability of the paper.
*RE: Thank you for the suggestion. This is to inform the reviewer that we have done the English*
*proof reading by professional editing service for journals.  We have moved the previous section*
*3.2. 1 to section 2. We have rechecked  and revised the references and overall organization of the*
*manuscript in order to improve the readability of the paper.*
Q2: Another concern is that whether the model assumption is valid for the study area. For example,
both the transect of the tsunami deposit sites and the reference line (Figure 6) is oblique to the
coastline, meanwhile the model assumes that the coordinate x for the forward simulation is
perpendicular to the shoreline (equation 2). I'm curious that how likely the direction of tsunami
inundation was consistent with these lines. Satellite imageries show that the geometry and directions
of the sandy ridges are quite complex, implying the tsunami inundation might have been affected
the local topography. Fujino et al. (2010) mentioned that the measurement of the flow direction was
not many and in fact only single measurement was made near the coastline of the transect. There
may be an uncertainty in the tsunami inundation direction (and probably the sediment source). If this
is the case, additional computation of the forward and DLNN model using reference line with
different directions are needed.
*RE: Thank you for the comment. The transect was considered along the flow direction the flow*
*direction and because of the complicated local topography the perfectly perpendicular locations*
*were not possible to collect. In the previous paper Mitra et al., 2020 it was presented that the DNN*
*inverse model works well perpendicular to the shoreline and Sendai plain was one of the most*
*well-preserved tsunami deposits from 2011 Tohoku-oki tsunami. Moreover, practically in recent*
*or ancient tsunami deposits, it is often difficult to obtain samples from perfectly shore*

*perpendicular transects due to anthropogenic disturbances or complicated topography. Comparison between the model estimates and the observed flow depths represents that even if we are unable to obtain completely shore-perpendicular transect, the DNN model still predicts reasonable and realistic flow conditions.*

*Q3* With regard to Section 4.3, I think the flow speed comparison is problematic, since the model assumes a constant flow speed over the inversion region and it is not clear whether it represents either an average, maximum or something else. This also applies to the measured flow speeds. Unless the attributes of the measured flow speeds (i.e. average, maximum or other) are specified, the measured values cannot be compared with inversion results. The comparison to the inversion results of the TsuSedMod also needs a careful discussion, since the TsuSedMod employs different model assumptions and formulations. It is not clear how the comparison of the two different inversion results are justified.

*RE: The flow velocity in this model has been considered as constant, and this reconstructed velocity should be compared with the spatially averaged velocity of the field observation. In Mitra et al. (2020), the result of velocity reconstruction was compared with the field measurements that varied in space. As a result, the estimated value matched the spatial average of the measurements. Our reconstructed values are the only estimates available for this region, and we do not consider that these existing velocity values validate our model because of the different regions where the measurements were taken. In addition, it is true that we do not know the timing of the velocity values that we referred in this paper. Nevertheless, we believe that referring to estimates of velocities in surrounding areas may be useful in interpreting whether the velocity of the tsunami inundation flow in this region was unique.*
*We have removed the comparison with the TsuSedMod as the model assumption and formulation of TsuSedMod is different from our model.*

Q3: The idea of coupling DLNN with other tsunami hydrodynamic model, such as the well validated TUNAMI-N2, is very interesting. Although it must be computationally expensive, the DLNN inversion can include much more physically plausible hydrodynamic models to improve the model performance. I suggest to expand on this aspect, such as outlining a road map and future challenges.

*RE: Thank you for the suggestion. We agree that our proposal might be computationally expensive. In our revised manuscript we have added few details about this. The modified text is as follows:*

*P. 22, line 370, In addition to the source model, this model also includes tsunami sediment transport calculation that consists of bed load layer and suspended load layer. However, the calculated value of the sediment thickness was overestimated as the assumption of movable bed for a large area caused excessive erosion of the ground (Masaya et al., 2019).*
*Line 379, The model calculation of Masaya et al. (2019) relies on the estimation of a single set of fault parameters, which were not widely explored to obtain the optimal parameters. In future, Model TUNAMI-N2can be potentially used as the forward model in DNN inverse model to*

*consider two-dimensional behavior of tsunamis. To do so, the model needs to be modified for*
*considering sediment transport of multiple grain size classes.*
**Minor comments:**

1. *Abstract:* I suggest to use 'distance' rather than 'length'.
*RE: Thank you for your suggestion. We decided to change the 'length' to 'distance' in the*
*entire document.*
2. P-1, Line 22, Out of context?
*RE: Thank you for the comment. We have removed the sentence at line 22 considering the*
*sentence as out of context.*
3. P-2, Line 25, I can understand what the authors intended to say, but the wording is unusual.
Consider to revise, line 28, I understnad what the authors intended to say, but the structure of the
sentence is not easy to follow. Consider to revise, line 31, I understnad what the authors intended
to say, but the structure of the sentence is not easy to follow. Consider to revise.
*RE: Thank you for the suggestion. We have revised the text as follows,*
*P.2, line 25, Indeed due to the lower tsunami risk and the higher return period of high*
*magnitude tsunamis (600 years) (Suppasri et al., 2015), the degree of preparedness, for*
*example, effective evacuation techniques, and appropriate awareness are still in the early*
*stageof development in Thailand (Suppasri et al., 2012).*
*Line 28, Suppasri et al. (2012) reported that, the nation has implemented post-tsunami*
*precautionary measures such as, the construction of evacuation shelters at a safe height and*
*distance from the coastline along with the evacuation routes with evacuation regulations,*
*memorial parks, appropriate structural design and land use management which were aimed*
*at dealing with tsunami waves.*
*Line 33, To propose further regulations for evacuation plan and tsunami hazard mitigation,*
*evaluating the extent of tsunamis with the, flow velocity and the maximum height that the*
*tsunamis could reach is important (Pignatelli et al., 2009).*
4. P-2, Line 25, lower?
*RE: In the sentence we tried to explain that the tsunami preparedness was not appropriate*
*because of higher tsunami return period that resulted into extensive damages.*
5. P-2, Line 36, The sentence is incomplete.

*RE: Thank you for identifying this. We considered revising the sentence as "Meanwhile, other flow parameters, such as flow velocity and depth, remain largely unknown".*

6. P-2, Line 40, I understnad what the authors intended to say, but the structure of the sentence is not easy to follow. Consider to revise.

   *RE: Thank you. We have considered revising the sentence.*
   *P.2, line 43, It is important to obtain the flow conditions essential to tsunami hazard mitigation in terms of devising future resilient structural measures by investigating tsunami deposits, which provide crucial information on the flow discharge and the extent of the tsunami inundation (Dawson and Shi, 2000; Udo et al., 2016; Sugawara and Goto, 2012;45Furusato and Tanaka, 2014; Sugawara et al., 2014; Koiwa et al., 2018; Masaya et al., 2019).*

7. P-2, Line 57, It seems some of the papers are not cited appropriately. For example, this paper is a report of forward tsunami propagation and inundation modeling. Check all other citations carefully whether they are used appropriately.

   *RE: Thank for the comment. We will recheck the citations and we revise the sentence as follows:*

   *P. 3, line 58, To reconstruct quantitative values of tsunami characteristics from the deposits, various numerical forward and inverse models which incorporate sediment dynamics, and transport and depositional equations have been established.*

8. P-3, Line 60, The structure is not easy to follow. Compare the estimated and observed values for the inundation distance and flow speed and depth.

   *RE: We agree with the reviewer. We added few more sentences to compare the estimated and observed values. These sentences are added*
   *Line 65 "The DNN inverse model predicted the tsunami flow conditions. The reconstructed inundation length was 4,045m which is close to the original maximum inundation length of approximately 4,020 m, values of run-up flow velocity were 5.4 m/s which was close to the spatial average of the measurements which ranged from 1.9 to 6.9 m/s, and the estimations of the inundation depth was 4.11 m which was also within the range of the in-situ measured values from Sendai plain."*

9. P-3, Line 65, The sentence is complicated. Consider to revise.

   *RE: Thank you for the comment. In response to your comment we have revised the text as follows:*
   *Line 74, "The Phra Thong island is one of the locations where the tsunami deposits were preserved without a great amount of topographic irregularities with almost no anthropogenic disturbances in the island."*

10. P-3, Line 67, This sentence must be divided into two sentences.

*RE: Thank you for the comment. In response to your comment we have revised the text as follows: Line 66, The coastlines of Phra Thong island were severely eroded and retreated by the 2004 tsunami. However, the presence of widespread mangrove forests with other waterborne plant debris helped in the identifications of the extent and direction of the flow.*

11. P-3, Line 75, Difficult to understand how the flow conditions can be used to estimate sediment characteristics.

*RE: Thank you for the comment. We added the following paragraph to modify the text,*

*P.3, Line 86, Here, we conduct an DNN inverse analysis of the tsunami deposits measured at Phra Thong island and reconstruct the flow conditions such as the maximum inundation length, flow velocity, maximum flow depth and sediment concentrations of five grain-size classes. The inverse model was based on the forward model, which was proposed by Naruse and Abe, (2017). The forward model calculations were iterated at random initial flow conditions to produce artificial training data sets that represent depositional characteristics such as the spatial distribution of thickness and grain-size composition. Using the artificial training datasets, the DNN was then trained to establish a relation between the depositional characteristics and the and the flow conditions. The post-trained DNN model was ready to predict flow conditions from the tsunami deposits after the performance of the trained DNN was verified using test data sets. The 1-D cubic interpolation was applied to the field data sets of Phra Thong island to fit the dataset in to model grids. Finally, this DNN inverse model was applied to the field data sets from the Phra Thong island, Thailand to reconstruct the flow conditions of 2004 Indian Ocean tsunami. We also used the reconstructed flow conditions to estimate the spatial distribution of the volume per unit area and grain-size composition from Phra Thong island and compare the distribution with the measured data. Our inverse model was already validated to be effective for 2011 Tohoku-oki tsunami deposits distributed in Sendai Plain. In case of Phra Thong island, we validated the results by the field measurements of the tsunami flow depth. Also, the estimated thickness and grain size distribution of tsunami deposits were compared with the actual measurements. Our inverse analysis results could be used for designing future tsunami hazard assessments and disaster mitigation strategies in Thailand.*

12. P-4, Line 102, Expression for the rate of total sedimentation is missing.

*RE: Thank you for identifying this. The typing error was corrected as equation 5 in the revised manuscript.*

13. P-4, Line 104, Consider to explain how likely the assumptions are valid for this case.

*RE: Thank you for your comment. The explanation is as follows:*

*P. 8, line 161, The velocity of the run-up flow of the tsunami, U is assumed as uniform and steady, but the inundation depth varies in time and space. Hence, this model simplification is called the quasi-steady flow assumption (Naruse and Abe, 2017).*

14. P-4, Line 113, Elaborate how the grain-size classes were determined. Measurement of tsunami deposit? Complete descriptions for the data (number of data points and their locations, thickness, grain-size distribution and other sedimentological features) are needed..

*RE: Thank for the suggestion. We have added a new figure (figure 3) on the thickness and grain size distribution and also moved the study area section before the methodology section.*

15. P-5, Line 119, Figure 1a must be placed at first. Consider to exchange the position of Figure 1a and 1b, and swap the names. Or just revise the structure of this section

*RE: Thank you for the suggestion. We have done the necessary changes with the position of the diagrams.*

16. P-5, Line 120, measured (or observed)

*RE: Thank you for the comment. These are all measured deposit. We have replaced the word "natural" with "measured".*

17. P-5, Line 132, Complete descriptions are needed for the basis for selection of the range of the input values. The description may include appropriate reference to field observations or experimental data.

*RE: Thank you for the comment. We have added the necessary references for the selection of the range of input values.*
*P. 10, line 202, The range of parameters adopted in this study is applicable to most of the large-scale tsunami-inundated areas as the ranges have been selected with several case studies of tsunamis that includes mostly field measurements ,survivor video and numerical analysis (Mori et al., 2011; Wijetunge, 2006; Szczuciński et al., 2012; Matsutomi and Okamoto,2010; Abe et al., 2012; Fritz et al., 2006; Nandasena et al., 2012; Goto et al., 2014).*

18. P-5, Line 146, The sentence is complicated. Consider to revise.

*RE: Thank you for the comment.*

*P. 11, line 215, To apply the inverse model to the measured values of field dataset from Phra Thong island in 1-D vectors, the collected datapoints must be fit into that fixed coordinate system of the model.*

19. P-7, Line 165, Difficult to understand what 'epochs' indicates. Iterations?

*RE: Thank you for your comment. The number of epochs indicates the number of times that a full data set has passed the optimization calculation. The specific number of epochs was determined based on the rates of the progress of the training. We can modify the text as follows:*

*P. 11, line 236, The training process proceeded with a certain number of epochs that indicates the iterations of the optimization calculation by the full dataset.*

20.  P-8, Line 173, Why the flow depth was biased?

*RE: Thank you for the comment. We have added the clarification in the discussion section.*
*P. 20, line 291, The bias was caused by the internal algorithm and neural network structure,*
*but we hope the biasness will be sorted if we improve the neural network structure in future.*
*In future studies, the algorithm of the neural network structure can be improved to eliminate*
*or reduce the bias of the parameter.*

21. P-8, Line 178, This section must be moved to somewhere in the section 2. Not in the result
section. Readers may want to see whether model assumptions for the sediment source is valid in
this island. Were the tsunami deposit along the transect totally brought from the sea?

*RE: Thank you for your suggestion. We decided to move the section before methodology.*

22. P-8, Line 180, Swap Figures 5 and 6. Figure 5 must be appeared first in the text. Give name of
insets (a, b and c) for current Figure 6 for the convenience of readers.

*RE: Thank you for the comment. We have revised the positions and labelling of the figures.*

23.  P-8, Line 191, However, the reference line seems not to be perpendicular to the coastline,
according to Figure 6. Explanations are needed regarding the compatibility to the model
assumption (i.e. inversion transect must be perpendicular to the coastline).

*RE: We are assuming that this comment is the same comment or related to Q2. Please refer*
*to the reply of Q2.*

24. P-8, Line 192, The sentence is complicated. Consider to revise, Line 204, The sentence is
complicated. Consider to revise.

*RE: Thank you for your comment. We have modified the text as follows:*
*P. 4, line 119, The sediment from shallow seafloors were transported and deposited in large*
*volumes of sand sheet deposition widely along the coast, with the deposit is largely composed*
*of medium to fine sand. The deposit became thinner and finer in a landward direction,*
*becoming very fine at the landward limit of the inundation.*
*P. 14, line 256, The jackknife standard error was calculated for different sampling window*
*sizes of the datasets. Figure 9 represents that the error decreased as the sampling window*
*was increased.*

25.  P-10, Line 206, How do we interpret the increasing trend for the maximum flow depth?

*RE: Thank you for your comment. Figure 9c shows that the jackknife standard error of*
*maximum flow depth has an increasing trend up to sampling window size 1500 m. After*

*sampling window size 1500 m, the error started to decrease and after 1600 m is was very low*
*yet stable.*

26. P-11, Line 212, The sentence is quite complicated and is not easy to follow in terms of English
writing. Consider to revise.

*RE: Thank you for your comment. We have revised the section of the paragraph as follows:*
*P. 15, line 264, The subsampling test demonstrated that the inversion model had a mean bias*
*of 10.82 m for maximum inundation distance (Figure 10) while the predicted result by DNN*
*was 1700 m. Likewise the predicted results for the flow velocity was 4.63 m/s and it was 4.82*
*m for the maximum flow depth, with the mean bias obtained from the subsampling results*
*being 0.14 m/s for flow velocity and -0.43 m for maximum flow depth, which were exactly in*
*line with the values obtained from the testing of the trained DNN model without the*
*subsampling test.*

27. P-11, Line 212, This figure must be appeared after current Figure 7.

*RE: Thank you for the suggestion. We have rearranged the figures.*

28. P-12, Figure 6, This is not a appropriate caption for this figure.

*RE: Thank you for the comment. We apologize for the typo in the caption. We have revised*
*the caption and labelling as follows:*

*P. 5, Figure 1. (a) Location of study area in southwestern Thailand. (b) Phra Thong island*
*and adjacent landmark areas where 2004 Indian ocean tsunami inundated. (c) Locations of*
*study sites at Phra Thong island. The 2004 tsunami inundated about 2 km inland.*

29. P-14, Line 234, Are there any reason for the biased results for the flow depth?

*RE: We assume that this comment is related to comment number 18. Please refer to the*
*comment 18.*

30. P-17, Line 264, Is this value incudes the bias of -0.38m?

*RE: Thank you for the comment. This value does not contain the additional bias -0.38 (now*
*revised -0.43 m). We have added the text in line 322.*

31. P-17, Line 267, Elaborate what is Tsuji and KSCOE- http://www.nda.ac.jp/~fujima/TMD/.

*RE: Thank you for the comment. To address this point, in the revised version we have added*
*the details and reference of the Tsuji and KSCOE group on P. 21, line 324-326.*
32. This URL is no more available.
*Thank you for identifying this. We have updated the URL link in the revised manuscript.*
33. P-17, Line 276, Considering the local topographical variations, topographic elevation is needed
for other sites to calculate and compare the flow depths.
*RE: Thank you for the suggestion. Previously we averaged the elevation of the area. To*
*address this point, we will add the ranges of elevation and the respective flow depths. The*
*revision will be as follows:*
*P., 21, line 335, The maximum and measured flow heights from Phra Thong island were*
*reported 7.1 m and 5.5 m respectively (http://www.nda.ac.jp/~fujima/TMD/). The*
*corresponding maximum and minimum values of elevation are 3.1 and 1.1 m respectively*
*(Jankaew et al., 2008, 2011; Brill et al., 2012b). Hence, the approximate estimate of measured*
*maximum flow depth is ranged from 2.4 m to 6.0 m. Considering the bias correction of 0.43*
*m, the reconstructed value of maximum flow depth (5.3 m) falls within the range of measured*
*maximum flow depth values.*

34. P-17, Line 283, I think flow speed comparison is problematic, since the model assumes that the
flow speed is constant over the inversion region and it is not clear the inversion result represents
either an average, maximum or something else. This also applies to the measured flow speeds.
Unless the observational condition is specified, the measured values cannot be compared with
inversion results. The question is that the attributes of the measured flow speeds (i.e. average,
maximum or other).
*RE: Thank you for the comment. We assume this comment is related to Q2. Please refer to*
*the reply of Q2.*

35. P-17, Line 286, Is this mean the tsunami flow speed was measured using video footages from
aircrafts? Rossetto et al. (2007) did not mention about that.
*RE: We apologize for mentioning aerial footage. Rossetto et al. (2007) mentioned about the*
*video footage only. We have corrected the statement in line 350 of the revised version.*

36. P-18, Line 288, TsuSedMod uses different model assumptions and formulations. It is not clear
how the comparison of the two different inversion results are justified.

*RE: We agree with the reviewer. We decided to remove the comparison with TsuSedMod.*

37. P-18, Line 306, Before mentioning this, outlines of Masaya et al. (2019) must be introduced. It
seems that the paper includes not only the tsunami hydrodynamic simulation but also sediment
transport simulations.

*RE: Thank you for the suggestion. In our revised manuscript, we have added the outlines of*
*the model (Masaya et al., 2019) including hydrodynamics and sediment transport.*
*P. 22, line 370, In addition to the source model, this model also includes tsunami sediment*
*transport calculation that consists of bed load layer and suspended load layer. However, the*
*calculated value of the sediment thickness was overestimated as the assumption of movable*
*bed for a large area caused excessive erosion of the ground (Masaya et al., 2019).*

38. P-18, Line 311, The idea is very interesting. Although it must be computationally expensive,
the DNN inversion can include much more physically plausible hydrodynamic models to
improve the performance. Better to expand on this, such as outlining a load map and future
challenges.

*RE: We thank the reviewer for appreciating the idea. Currently, the model calculation of*
*Masaya et al., 2019 based on the single assumption of measured parameters which are not*
*optimized. Model TUNAMI-N2 could be integrated with the forward model of DNN inverse*
*model but the model needs to be modified for multiple grain size classes with the optimized*
*parameters. We have added these details in our revised manuscript as follows:*
*P. 22, Line 379, The model calculation of Masaya et al. (2019) relies on the estimation of a*
*single set of fault parameters, which were not widely explored to obtain the optimal parameters.*
*In future, Model TUNAMI-N2can be potentially used as the forward model in DNN inverse*
*model to consider two-dimensional behavior of tsunamis. To do so, the model needs to be*
*modified for considering sediment transport of multiple grain size classes.*